# Local oceanic $CO_2$ outgassing triggered by terrestrial carbon fluxes during deglacial flooding

Thomas Extier[1,a], Katharina D. Six[1], Bo Liu[1], Hanna Paulsen[1], and Tatiana Ilyina[1]

[1]Max Planck Institute for Meteorology, Hamburg, Germany
[a]now at: EPOC, UMR 5805, CNRS, Université de Bordeaux, Pessac, France

**Correspondence to:** Thomas Extier (thomas.extier@mpimet.mpg.de)

**Abstract.**

Exchanges of carbon between the ocean and the atmosphere are key processes that influence past climates via glacial/interglacial variations of the $CO_2$ concentration. The melting of ice sheets during deglaciations induces a sea level rise which leads to the flooding of coastal land areas, resulting in the transfer of terrestrial organic matter to the ocean. However, the consequences of such fluxes on the ocean biogeochemical cycle and uptake/release of $CO_2$ are poorly constrained. Moreover, this potentially important exchange of carbon at the land-sea interface is not represented in most Earth System Models. We present here the implementation of terrestrial organic matter fluxes into the ocean at the transiently changing land-sea interface in the Max Planck Institute for Meteorology Earth System Model (MPI-ESM) and investigate their effect on the biogeochemistry during the last deglaciation. Our results show that during the deglaciation, most of the terrestrial organic matter inputs to the ocean occurs during Meltwater Pulse 1a (between 15-14 ka) which leads to the transfer of 21.2 GtC of terrestrial carbon (mostly originating from wood and humus) to the ocean. Although this additional organic matter input is relatively small in comparison to the global ocean inventory (0.06 %) and thus doesn't have an impact on the global $CO_2$ flux, the terrestrial organic matter fluxes initiate oceanic outgassing at regional hotspots like in Indonesia for a few hundred years. Finally, sensitivity experiments highlight that terrestrial organic matter fluxes are the drivers of oceanic outgassing in flooded coastal regions during Meltwater Pulse 1a. Furthermore, the magnitude of outgassing is rather insensitive to higher carbon to nutrients ratios of the terrestrial organic matter. Our results provide a first estimate of the importance of terrestrial organic matter fluxes in a transient deglaciation simulation. Moreover, our model development is an important step towards a fully coupled carbon cycle in an Earth System Model applicable for simulations at glacial/interglacial cycles.

## 1 Introduction

Since the middle to late Pliocene (approximately 3 million years ago) the climate has undergone large variations including glacial and interglacial periods, associated with changes in ice sheets volume, sea level, oceanic circulation and atmospheric $CO_2$ concentration (e.g. Kohfeld and Ridgwell, 2009; Sosdian and Rosenthal, 2009; Spratt and Lisiecki, 2016). The last deglaciation defined to be from 21 to 10 ka (thousands of years before present) is the most recent manifestation of these changes and has been extensively examined in studies based on proxy records (e.g. Barker et al., 2009; Denton et al., 2010;

Clark et al., 2012; Marcott et al., 2014) and Earth System Models (ESMs) with a focus on the timing of the last deglaciation, on the oceanic thermohaline circulation or on the ice sheet dynamics (e.g. Bonelli et al., 2009; Liu et al., 2009; Menviel et al., 2011; Roche et al., 2011; Heinemann et al., 2014; Ivanovic et al., 2016; Klockmann et al., 2016). The melting of ice sheets during the last deglaciation is accompanied by a sea level increase of about 95 m (Lambeck et al., 2014; Spratt and Lisiecki, 2016), resulting in flooding of land coastal areas and changes in the oceanic coastlines. In this case, the carbon and nutrients previously stored on land in vegetation and soil are transferred to the ocean, potentially impacting the global ocean biogeo-
chemistry with implications for the uptake and release of carbon by the ocean. Indeed, the carbon and nutrients bound in the terrestrial material might change the ocean biogeochemistry once they have decomposed. A specificity of this terrestrial organic matter (terrOM) is the higher carbon to nutrients ratio than the marine organic matter. But as of now, the role of this terrestrial organic matter transfer to shelf areas on the glacial/interglacial atmospheric $CO_2$ variations remains poorly constrained. The
representation of these fluxes does not exist in climate models, making a consistent quantification of their role in the context of the global carbon cycle challenging. The $CO_2$ increase from 188 to 264 ppm observed during the last deglaciation (Bereiter et al., 2015) results from the combination of mechanisms partly associated with ocean outgassing, following changes in the terrestrial and marine carbon cycle and could be directly dependent of the land-sea organic matter fluxes during flooding events.

Previous attempts have tried to explain the glacial/interglacial changes in atmospheric $CO_2$ concentration from an ocean perspective using Earth System Models of Intermediate Complexity (EMICs) or paleoproxy records. For example, Schmittner et al. (2007) used a coupled climate-carbon cycle model to show that the oceanic carbon content decreases following the AMOC shutdown, explained by the reduced ocean solubility due to higher Southern Ocean temperature. Menviel et al. (2008) found from glacial water hosing simulations that after a small increase in atmospheric $CO_2$, the carbon oceanic sink becomes
dominant leading to a decrease of atmospheric $CO_2$ of ∼10 ppm, also consistent with the response of a coupled climate–carbon cycle model to freshwater discharge (Obata, 2007). More recent work from Menviel et al. (2014) combined paleodata and climate simulations from two EMICs (LOVECLIM and UVic ESCM) to show that the ocean can act either as a sink or a source of carbon depending on the bottom water transport in the Pacific Ocean. But, changes in the physical conditions of the ocean are not the only processes that could explain the glacial/interglacial atmospheric $CO_2$ variations. Sigman et al. (2010) have shown
that a reduced oceanic biological production has the effect to increase the atmospheric $CO_2$. A weakening of the biological pump efficiency (resulting in a decrease of the oceanic alkalinity), could be the driver of the atmospheric $CO_2$ rise. Other processes like a change of the oceanic phosphate inventory or a decrease of the remineralization length scale of particular organic matter can also increase the atmospheric $CO_2$ concentration by several ppm (Menviel et al., 2012).

Part of the atmospheric $CO_2$ increase from a glacial to an interglacial period can also be explained by the increase of the
land carbon inventory to compensate for the outgassing of the ocean. Isotopic measurements of $\delta^{18}O$ of atmospheric $O_2$ suggest an increase in land carbon of around 330 GtC between the Last Glacial Maximum (LGM) and the Preindustrial (Ciais et al., 2012). Recent modelling estimates also indicate that the land carbon content was lower of 450 to 1250 GtC during the LGM than during the Preindustrial (e.g. Menviel et al., 2011; Jeltsch-Thömmes et al., 2019; Müller and Joos, 2020) due to colder and drier climate conditions and to the presence of large ice sheets. However, the exact mechanisms responsible for the

glacial/interglacial atmospheric $CO_2$ changes remain unclear and all the previous modelling attempts to explain these variations only considered the ocean-atmosphere $CO_2$ gas exchange and carbon fluxes between the land carbon pool and the atmosphere, ignoring direct carbon fluxes across the land-sea continuum and the associated consequences on the marine biogeochemistry.

Direct carbon fluxes between land and ocean can result from flooding of coastal areas. During the last deglaciation, several short-term events of a rapid sea level rise are observed. They are referred to as meltwater pulse events, following the melting of the ice sheets, and have consequences on the oceanic circulation, biogeochemistry and climate (e.g. Weaver et al., 2003; Stanford et al., 2006). One of this event, the Meltwater Pulse 1a (MWP1a) around 14.65 ka, is associated with a rapid sea level increase from 8.6 to 20.2 m within 500 years (Deschamps et al., 2012; Liu et al., 2016; Lin et al., 2021). The source of the large amounts of meltwater is, yet, unclear. Either a partial melting of the Northern Hemisphere Ice Sheets or of the Antarctic Ice Sheet, or changes of both ice sheets are currently investigated by modelling studies (Mackintosh et al., 2011; Golledge et al., 2014; Weber et al., 2014; Gomez et al., 2015; Gregoire et al., 2016; Yeung et al., 2019). However, due to the rapid sea level change, MWP1a is a particularly interesting period to look at the role of terrestrial organic matter fluxes on atmospheric $CO_2$ variations. In this study, the freshwater inputs of MWP1a to the ocean are deduced from a prescribed ice sheet reconstruction and are located in the Northern Hemisphere (Tarasov et al., 2012). We are aware of the uncertainty of the origin of this meltwater pulse which could potentially impact the location of flooded land coastal areas, and ultimately the land-sea fluxes.

We present here for the first time a coupled transient simulation over the last deglaciation using MPI-ESM (Max Planck Institute Earth System Model) that takes into account (1) a fully interactive adaptation of the ocean bathymetry with corresponding changes of the land-sea distribution (Meccia and Mikolajewicz, 2018), (2) a transient river routing (Riddick et al., 2018) and the representation of new processes that are (3) the automatic adjustment of marine biogeochemical tracers under changing ocean bathymetry and land-sea distribution and (4) the fluxes of terrestrial organic matter between land and ocean during flooding events. A special focus is placed on the time period of the large freshwater inputs between 15-14 ka corresponding to Meltwater Pulse 1a to address the role of terrestrial organic matter fluxes on the $CO_2$ fluxes between the ocean and the atmosphere in context of this millennial event.

## 2 Model description

MPI-ESM is a comprehensive Earth system model composed of four components and one coupler: the atmospheric component ECHAM6.3, the ocean dynamics component MPIOM1.6, the ocean biogeochemistry component HAMOCC6, the land, hydrology and dynamic vegetation component JSBACH3.2 and the coupler OASIS3-MCT. It is currently used in the version 1.2 following the latest updates detailed in Mauritsen et al. (2019). Several components of MPI-ESM have been further extended with new developments to take into account a varying land-sea mask during period of sea level changes (Meccia and Mikolajewicz, 2018) and the river routing (Riddick et al., 2018), as described in the following sections. In addition, we present here

new model developments that take into account the terrestrial organic matter fluxes between land and ocean at a transiently changing land-sea interface as well as their effect on the ocean biogeochemistry.

## 2.1 HAMOCC

The global ocean biogeochemical model HAMOCC (HAMburg Ocean Carbon Cycle) is part of the ocean component MPIOM (Max Planck Institute Ocean Model) which is used to simulate the oceanic physics by resolving primitive equations under the hydrostatic and Boussinesq approximation on a C-grid with a free surface for every time step (1h) (Jungclaus et al., 2013). HAMOCC simulates the biocheochemical tracers and processes in the oceanic water column, the sediment and at the air-sea interface. HAMOCC has been previously described in Ilyina et al. (2013) and then revised in Mauritsen et al. (2019) with

additional processes like marine nitrogen fixation by Cyanobacteria (Paulsen et al., 2017) and nitrogen deposition. Recently, additional ocean tracers have been added in HAMOCC to simulate carbon isotopes (Liu et al., 2021). In this paper we use the HAMOCC6 version as described in Mauritsen et al. (2019) with specific new features for the coupling of the land-sea-continuum that are described in the next sections. The grid configuration of HAMOCC is identical to the ocean component MPIOM and consists of a bipolar grid with one pole over Greenland and another pole over Antarctica. The coarse-resolution

ocean grid used for paleoclimate purpose is noted GR30 and has a horizontal resolution of about 300 km. The vertical resolution is composed of 40 unevenly spaced layers with level thickness increasing with depth. Details on the coupling between MPIOM and HAMOCC can be found in Maier-Reimer et al. (2005).

The biogeochemistry of the water column is computed based on the extended NPZD (Nutrients, Phytoplankton, Zooplankton

and Detritus) model described in Six and Maier-Reimer (1996) and includes the following pools: phytoplankton, zooplankton, cyanobacteria, dissolved organic matter, particulate organic matter, dissolved inorganic phosphate, dissolved inorganic nitrate, dissolved iron, $O_2$, dissolved silicate, opal, calcium carbonate and nitrous oxide ($N_2O$). The model solves the ocean carbonate system (mocsy 2.0) for total dissolved inorganic carbon and total alkalinity according to the protocol of the Ocean Model Intercomparison Project (Orr and Epitalon, 2015). We also use the total pH scale and equilibrium constants recommended by

Dickson et al. (2007) and Dickson (2010). All tracers are mass conserving and follow the variations of temperature and salinity with respect to hydrodynamical processes. In contrast to Mauritsen et al. (2019), we revised the flux of particulate organic matter. Instead of using a linear increasing sinking speed we use a constant sinking speed of $5\,\mathrm{m\,day^{-1}}$ and introduce a temperature dependent remineralisation rate with a Q10 of 2.3. The remineralization of organic matter can be aerobic or anaerobic with denitrification or sulfate reduction when $O_2$ concentration passes a critical threshold. This version of HAMOCC also uses an

updated procedure to account for weathering fluxes entering the ocean. Instead of a globally uniform distribution, weathering fluxes are only distributed along the coast. The water column interacts with atmospheric components by air-sea exchange of gaseous tracers ($O_2$, $N_2$, $N_2O$ and $CO_2$). All atmospheric concentrations are prescribed. Dust deposition from the atmosphere to the ocean is also taken into account based on the time slice estimates of Albani et al. (2016) (see details in Section 2.4).


The simulation of the oceanic sediment follows the approach of Heinze and Maier-Reimer (1999). The biogeochemical tracers of the sediment include dissolved inorganic carbon, alkalinity as well as dissolved phosphate, nitrate, silicate, oxygen, iron, $N_2$ and $H_2S$ for the pore water fraction and detritus, opal, calcium carbonate and clay for the solid fraction. The processes of decomposition of detritus, dissolution of opal and calcium carbonate and the carbon chemistry are similar to those in the water column. There is vertical diffusion of dissolved tracers in pore water within the sediment column, as well as a diffusive exchange with the water column above. The sediment is resolved by 12 layers with increasing thickness and decreasing porosity from top to bottom, the bottom layer being defined as an underlying burial layer. In the burial, no biogeochemical processes are active and there is no pore water diffusion between burial and active sediment layers. This HAMOCC version also includes erosion under higher bottom shear stress. At most half of the first sediment layer volume of the solid components can be eroded within one time step. Due to remineralization and dissolution of solid components as well as erosion, gaps might be produced within the sediment layers. They are removed by shifting the solid sediment constituents downward within the sediment column. To guarantee that each sediment layer is properly filled with solid components according to solid fraction, the sediment column is shifted upward and filled with sand from the burial layer. Sand is an inert component and can not be eroded (Mathis et al., 2019).

## 2.2 Dynamic land-sea mask and hydrological discharge

In context of ice sheets growth or decay and associated changes in freshwater input and sea level, it is necessary to take into account the variations in ocean bathymetry and coastlines when performing transient simulations. Meccia and Mikolajewicz (2018) describe a new automatic method for interactive bathymetry and land-sea mask changes in MPIOM, allowing simulations for transient time periods. To do so, an ice sheet reconstruction is used to compute the time-dependent freshwater fluxes to the ocean and to derive the topography, which is then used to obtain changes in bathymetry and land-sea mask. Changes in ocean bathymetry and the land-sea mask are updated every 10 years. In the case of ocean expansion, new grid cells are flooded by the surrounding water. For HAMOCC, this also includes pore water of new sediment grid cells. The solid fraction of the new sediment column is filled with sand. This automatic bathymetry adjustment conserves mass (i.e. salt content and all other oceanic tracers) at the global and regional scales (changes in ocean volume match the freshwater input and the global inventory of tracers is constant in the absence of sources or sinks).

Changes in ice sheets volume since the Last Glacial Maximum induce changes in the river pathways that need to be taken into account to properly estimate the runoff during the last deglaciation. Riddick et al. (2018) presented new developments within the land surface component JSBACH (Jena Scheme for Biosphere-Atmosphere Coupling in Hamburg) (Reick et al., 2013; Kleinen et al., 2020; Reick et al., 2021) and the Hydrological Discharge model (Hagemann and Dümenil, 1997; Hagemann and Gates, 2001) to automatically generate dynamic river directions and flow parameters for past conditions. First the orography is adjusted and corrected. Then river directions and flow parameters are generated and the location of the river mouths are determined. The ice sheet height and isostatic adjustments are taken from an ice sheet reconstruction and the land-sea mask is generated using the technique described above. These changes in freshwater discharge to the ocean have significant impacts

on the global oceanic circulation (via the North Atlantic and Arctic Oceans) and as a consequence also affect atmospheric circulation.

## 2.3 Land-sea carbon and nutrients transfer

In this version of MPI-ESM, we account for the fact that coastal areas are flooded due to increasing sea level (following the melting of the ice sheets). ECHAM6.3 does not include fractional grid boxes, i.e. a single grid box is treated as land (ocean) as
long as the land (ocean) fraction is larger than 50 %. Flooding, i.e. conversion from land to ocean, occurs if the land fraction within one grid cell is less than 50 %. Drying of an ocean grid cell in the case of a decrease of the sea level is also considered. In this case, all pore water tracers, i.e. phosphate, nitrate, dissolved inorganic carbon and alkalinity are redistributed to the water column to guarantee mass conservation and the solid parts of the sediment are considered as inactive land pools.

In the land component JSBACH, each grid box represents the diversity of plant functional types (PFTs) depicting various veg-
etation types (Reick et al., 2021). 11 PFTs are described in the model: 8 for natural vegetation and 3 for land-use types (which are not used in this deglaciation simulation). The land module also accounts for coupled carbon and hydrological cycles. As part of the atmospheric component, the land component JSBACH uses the same coarse-resolution grid noted T31 with an approximate grid spacing of 400 km (different from the GR30 grid of MPIOM). In the case of flooding, the terrestrial organic matter of the flooded grid cell is collected on the T31 grid and remapped to the GR30 grid. The transfer from land to ocean is
mass conserving.

As a consequence of flooding events, the land carbon pools (wood, woody litter above and below ground, humus) are transferred to corresponding oceanic pools in the water column and sediment (Figure 1). So in addition to the 25 prognostic tracers already existing in HAMOCC, we added 3 new prognostic tracers to represent wood, woody litter and humus. In case of flood-
ing, woody litter above ground is distributed uniformly over the water column of the corresponding ocean grid cells. Woody litter below ground and humus are transferred to the sediment where they participate in all sedimentary processes and can be eroded (Figure 1). Wood is located at the water-sediment interface. It is not advected and does not participate in erosion. All terrestrial organic particles in the water column sinks with the same speed as marine particulate organic matter. All terrestrial organic matter is remineralized using oxygen in case of aerobic remineralization or $NO_3$ and $NO_2$ if anaerobic remineralization
takes place (see equations below). The prescribed remineralization timescales in water are 100 years for wood, 10 years for woody litter and 5 years for humus. It is crucial to note that the stoichiometry of terrestrial organic matter differs from that of marine organic matter defined as C:N:P = 122:16:1 (Takahashi et al., 1985). The carbon to nutrients ratio for woody litter (above and below ground) is defined as C:N:P = 7600:51:1 (Goll et al., 2012). The carbon to nutrients ratio for wood is set to C:N:P = 3650:11:1 (Goll et al., 2012). The carbon to nutrients ratio for humus is C:N:P = 465:10:1 (Goll et al., 2012). Rem-
ineralization of terrestrial organic matter leads to changes in dissolved inorganic carbon, phosphate, nitrate, dissolved oxygen and alkalinity in the water column or the sediment. In contrast to the long-living terrestrial material that is transferred to the ocean, short-living material from the vegetation (green biomass, non-woody litter above and below ground) is recycled within one year. We assume conversion of land to ocean by flooding happens over 10 years and after this time, all the short-living

terrestrial organic matter is remineralized and emitted as $CO_2$ to the atmosphere (Figure 1). However, since we used prescribed

$CO_2$ concentrations in this simulation, the flooding induced terrestrial carbon emitted to the atmosphere has no effect on the climate.

**Pre-flooding conditions**          **Post-flooding conditions**

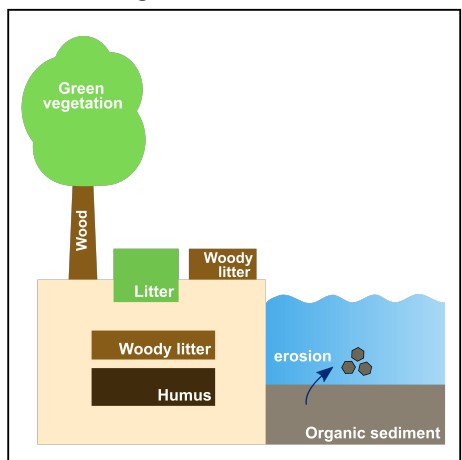
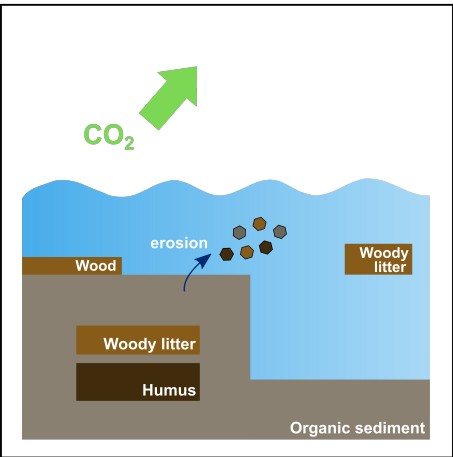

**Figure 1.** Scheme of the pre- and post-flooding environments for terrestrial organic matter.

Assuming that terrestrial carbon compounds that are carbohydrates $(CH_2O)_x$ give the classical organic matter composition defined as $(CH_2O)_x(NH_3^+)_yH_3PO_4$, we use the equations of Paulmier et al. (2009) presented below to calculate the oxygen and nitrate demand for remineralization, as well as alkalinity change. Considering that land biomass contains no excess $H^+$

compared to organic material in the ocean we obtain:

$$
\begin{aligned}
c &= a + 4 && \text{for oxygen} \\
z &= 0 = b - 2c - d + 5 && \text{for } H^+
\end{aligned}
\tag{1}
$$

We use the equation for aerobic remineralization with an autotrophic shortcut to nitrate as defined in Paulmier et al. (2009):

$$
C_aH_bO_cN_dP + (a + \tfrac{1}{4}b - \tfrac{1}{2}c + \tfrac{5}{4}d + \tfrac{5}{4})O_2 \rightarrow aCO_2 + dHNO_3 + H_3PO_4 + (\tfrac{1}{2}b - \tfrac{1}{2}d - \tfrac{3}{2})H_2O
$$

$$\tag{2}$$

The change in alkalinity is given by (-d-1) following Eq. (2). For complete denitrification (with conversion of the ammonium produced by anaerobic remineralization into $N_2$) we use the following equation (Paulmier et al., 2009):

$$C_a\text{H}_b\text{O}_c\text{N}_d\text{P} + (\tfrac{4}{5}a + \tfrac{1}{5}b - \tfrac{2}{5}c + 1)\text{HNO}_3 \rightarrow a\text{CO}_2 + \text{H}_3\text{PO}_4 + (\tfrac{2}{5}a + \tfrac{3}{5}b - \tfrac{1}{5}c - 1)\text{H}_2\text{O} + (\tfrac{2}{5}a + \tfrac{1}{10}b - \tfrac{1}{5}c + \tfrac{1}{2}d + \tfrac{1}{2})\text{N}_2$$

(3)

For the $\text{NO}_3^-$ change, the value corresponds to (alkalinity change + 1), i.e. $(\tfrac{4}{5}a + \tfrac{1}{5}b - \tfrac{2}{5}c + 1)$. This leads to the overall values presented in Table 1 for the different carbon and nutrients compositions of terrestrial organic matter.

| Terrestrial OM | C:N:P | Aerobic remineralization | | Anaerobic remineralization | |
| --- | --- | --- | --- | --- | --- |
| | | $O_2$ demand | $\Delta$Alk | $\text{NO}_3^-$ demand | $\Delta$Alk |
| Wood | 3650:11:1 | 3672 | -12 | 2926.6 | 2925.6 |
| Woody litter | 7600:51:1 | 7702 | -52 | 6110.6 | 6109.6 |
| Humus | 465:10:1 | 485 | -11 | 378 | 377 |

**Table 1.** Compositions of terrestrial organic matter with consumption of oxygen, nitrate and change in alkalinity during remineralization.

## 2.4 Model and experiment setup

Two spin-up runs (between 26-24 ka and 24-21 ka) were first performed to bring the physical, climatic and biogeochemical systems into quasi-equilibrium. We then ran the deglaciation simulation starting at 21 ka. The initial conditions and forcing are described next. The greenhouse gases concentrations, i.e. $CO_2$, $CH_4$ and $NO_2$ are from Köhler et al. (2017) and the orbital parameters are taken from Berger and Loutre (1991). The atmospheric $CO_2$ concentration is prescribed during these simulations to test and validate the state of the model with the new developments presented in previous section. A new simulation with prognostic atmospheric $CO_2$ will be performed in the future after additional model development to address the gap on the interaction between the ocean biogeochemistry and the climate during the last deglaciation. The model is forced with the ice sheet reconstruction GLAC-1D (Tarasov et al., 2012) and dust deposition from Albani et al. (2016) which is linearly interpolated over the deglaciation since the data is only available for specific time-slices (21 ka, 16 ka, 10 ka, 8 ka, 6 ka, 4 ka, 2 ka). The initial marine nutrients and carbon inventories are set to similar values to those of the present day ocean with global surface mean concentrations of 4.2 mmol N m$^{-3}$ for nitrate, 0.52 mmol P m$^{-3}$ for phosphate, 9.2 mmol Si m$^{-3}$ for silicate and 1915 mmol C m$^{-3}$ for DIC. We didn't adjust the nutrient concentrations for the 3.5 % change in the oceanic volume between the present day and the LGM. The global mean alkalinity concentration is set higher than present-day with 2300 mmol m$^{-3}$ to stabilize the marine chemistry to the lower LGM atmospheric $CO_2$ concentration. Weathering fluxes are set to a global constant value for the entire simulation to compensate for burial fluxes, based on the previous spin-up runs. The global weathering rates used to calculate the coastal input fluxes correspond to 1 kmol P s$^{-1}$ for organic material, 1720 kmol C s$^{-1}$ for inorganic material and 210 kmol Si s$^{-1}$ for silicate. The weathering fluxes are independent from the river discharge even if they might locally show small variations because the length of the coastline varies. Moreover, we consider that high latitudes coastlines

get only 20 % of the global value to avoid excessive inputs of silicate that would occur if the weathering rates were distributed uniformly over the coastlines (Lacroix et al., 2020). Finally, we assume that sea-ice contains only pure water and thus does not carry any biogeochemical tracer or salt. The land-sea mask, bathymetry, river directions and flow parameters are updated every 10 years.

Different sensitivity experiments have also been performed during Meltwater Pulse 1a by branching off new simulations at 15
ka from the deglaciation simulation. These sensitivity experiments ran for 1000 years. One sensitivity experiment has been run without terrestrial organic matter fluxes to the ocean to investigate the changes in sea-air carbon flux. A set of two sensitivity experiments has been run with modified carbon to nutrients ratios for terrestrial organic matter entering the ocean (details in Section 3.3). Furthermore, one sensitivity experiment has been run with modified stoichiometry and remineralization rates for terrestrial organic matter.

## 240 3 Results and Discussion

### 3.1 Ocean and land responses over the deglaciation

The last deglaciation simulation is characterized by several changes in the oceanic physics and biogeochemistry, as well as in the land carbon content. The first part of the deglaciation, i.e. 21-15 ka, is marked by small changes in the global ocean primary production from 52.1 GtC $y^{-1}$ to 52.6 GtC $y^{-1}$ (Figure 2f), as well as in the ocean physics with a relatively constant AMOC
(Atlantic Meridional Overturning Circulation) (Figure 2c). The LGM AMOC strength is 22.5 Sv (1 Sv = $10^6$ $m^3$ $s^{-1}$) which is within the range of a multimodel mean LGM maximum AMOC value of $23 \pm 3$ Sv (Muglia and Schmittner, 2015), even if it is still unclear whether the AMOC was weaker or stronger during the LGM than in the Preindustrial. One data assimilation study supports a strong AMOC during the LGM with a value of 21.3 Sv (Kurahashi-Nakamura et al., 2017). In contrast, a recent estimate based on modelling experiments constrained by isotopic data suggests a weaker AMOC during the LGM, with
values between 6 and 9 Sv (Muglia et al., 2018). In our model, the physical state of the ocean, and in particular the AMOC and the ventilation of the Southern Ocean, show only little variation before 15 ka. Thus, the global net air-sea $CO_2$ flux is generally close to zero until 17.3 ka (Figure 2e), and then becomes mostly negative, i.e. the global ocean becomes a carbon sink, for several millennia due to the prescribed rising atmospheric $CO_2$ mixing ratio (Figure 2d). We do not find an enhanced outgassing of $CO_2$ in the Southern Ocean due to an increased ventilation in our model. Sensitivity studies with an Earth System
Model of Intermediate Complexity suggested that the observed atmospheric $CO_2$ increase after 17.3 ka could be attributed to an enhanced formation of Antarctic intermediate and/or deep water due to decreased buoyancy forces and/or changes in the westerlies in the Southern Hemisphere (Menviel et al., 2018).

A rapid change is observed in the ocean between 15-14 ka with the global primary production decreasing by 4.94 GtC $y^{-1}$ and the ocean uptaking up to 0.30 GtC $y^{-1}$ (Figure 2e-f). This abrupt event is correlated to the large freshwater input of 0.5 Sv
originating from the North Atlantic and Arctic Oceans, called Meltwater Pulse 1a. As a result, the AMOC strength decreases from 20 to 3 Sv (Figure 2c). The mean Sea Surface Temperature (SST) decreases by more than 5 °C in the North Atlantic and the mean Sea Surface Salinity (SSS) decreases by around 3 psu. Globally, the mean SST and SSS decrease by 0.53 °C

and 0.24 psu, respectively. This decrease in AMOC streamfunction is also observed when looking at a cross-section in the Atlantic Ocean between the Last Glacial Maximum state and the minimum of the streamfunction at 14.5 ka. During the LGM,

the maximum of the streamfunction in the Atlantic Ocean is around 1200 m depth at 30° N (Figure 3). Between 15-14.5 ka the AMOC strength decreases significantly with only weak circulation remaining for the top 2000 m, the most important export being in the South Atlantic. After this large meltwater pulse and the AMOC streamfunction minimum at 14.5 ka, the Atlantic circulation takes 500 years to return to its original state. The variability of the simulated AMOC is mainly regulated by temporal changes in the volume of the prescribed ice sheet reconstruction. In our model, the ice sheet volume decrease is

considered as liquid meltwater discharge, ignoring the discharge of icebergs. Freshwater inputs deduced from the GLAC-1D ice sheet reconstruction show low variations during the LGM and only a slight increase during the Heinrich Stadial 1. Thus, we can't expect pronounced AMOC changes during the period of Heinrich Stadial 1 in the model. Between 15-14 ka, we simulate a decrease of the AMOC strength following massive freshwater inputs in the Northern Hemisphere. This period of MWP1a is also characterized by a rapid sea level increase, which is recorded in radiocarbon dates from the Sunda Shelf and U/Th

measurements on corals offshore from Tahiti, confirming a timing of MWP1a between 14.65 to 14.31 ka (Hanebuth et al., 2000; Deschamps et al., 2012). However, the temporal variation of the AMOC strength estimated from $^{231}$Pa/$^{230}$Th tends to show already a decrease between 17.5-15 ka, i.e. before the MWP1a, and an increase back to a high value between 15-14 ka (McManus et al., 2004). To achieve a good agreement between simulated and proxy-data derived AMOC variations, He (2011) showed in a modelling study the necessity of meltwater fluxes from Antarctica during MWP1a. Other hosing experiments also

emphasize the sensitivity of the oceanic circulation, and thus the AMOC, on the location of the freshwater input (e.g. Smith and Gregory, 2009; Menviel et al., 2011). As previously discussed, the meltwater input deduced from GLAD-1D is located primarily in the Northern Hemisphere, which might explain the different temporal evolution of the simulated AMOC. In the following, we refer to MWP1a as the period of rapid sea level change due to large freshwater inputs to the ocean to evaluate the effect of land-sea exchanges during flooding events.

The sea level change over the last deglaciation can be estimated based on the freshwater inputs to the global ocean, that are set accordingly to the GLAC-1D ice sheet reconstruction. Considering that the sea level at 21 ka is similar to the estimated one from Spratt and Lisiecki (2016), i.e. -120 m relative to present day, we present on Fig. 2b the modelled deglacial sea level estimate. Between 21-12 ka, the sea level in the model increases by 67.4 m, which is close to the estimate of 69 m from Spratt and Lisiecki (2016). During MWP1a, the global sea level change in the model shows quantitative differences compared

to Spratt and Lisiecki (2016) record. Uncertainties exist in the prescribed ice sheet reconstructions that could explain such difference. For instance, the ice sheet volume and the timing of freshwater input show noticeable differences between GLAC-1D and ICE-6G reconstructions (see Ivanovic et al., 2016 for a comparison). The global sea level increases of 19.6 m for the 500 years of large freshwater inputs in the model (Figure 2a,b). This is in the high range of the previous estimations with a global sea level increase from 8.6 to 20.2 m (Deschamps et al., 2012; Liu et al., 2016; Lin et al., 2021). Then between 14-12

ka, the sea level in the model only slightly increases in comparison to the Spratt and Lisiecki (2016) record.

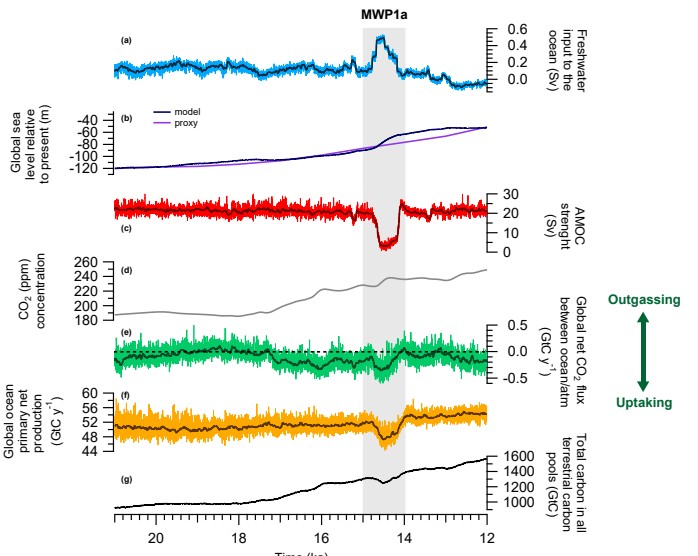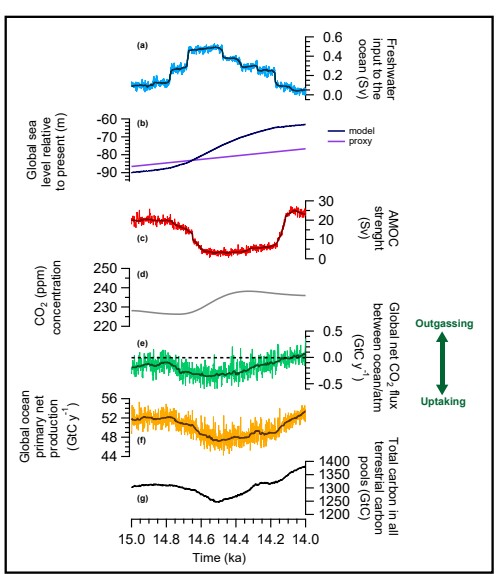

**Figure 2.** Time series of land, ocean and atmosphere variables over the last deglaciation. The presented outputs start at 21 ka. (a) Freshwater input to the global ocean. (b) Global sea level estimate from Spratt and Lisiecki (2016) (light purple) and modelled in MPI-ESM based on the freshwater inputs (dark purple). (c) Atlantic Meridional Overturning Circulation streamfunction. (d) $CO_2$ concentration measured in ice cores and prescribed in the model (Köhler et al., 2017). (e) Modelled global net $CO_2$ flux between the ocean and the atmosphere. Positive $CO_2$ flux means that the ocean is outgassing to the atmosphere and negative $CO_2$ flux means that the ocean is uptaking carbon. (f) Global ocean net primary production. (g) Total carbon in all terrestrial carbon pools, i.e. vegetation, soil and litter. The thick darker curves are 500 years running mean for the panel (a) and 50 years running mean for the panels (c), (e) and (f). A zoom over MWP1a is presented on the right.

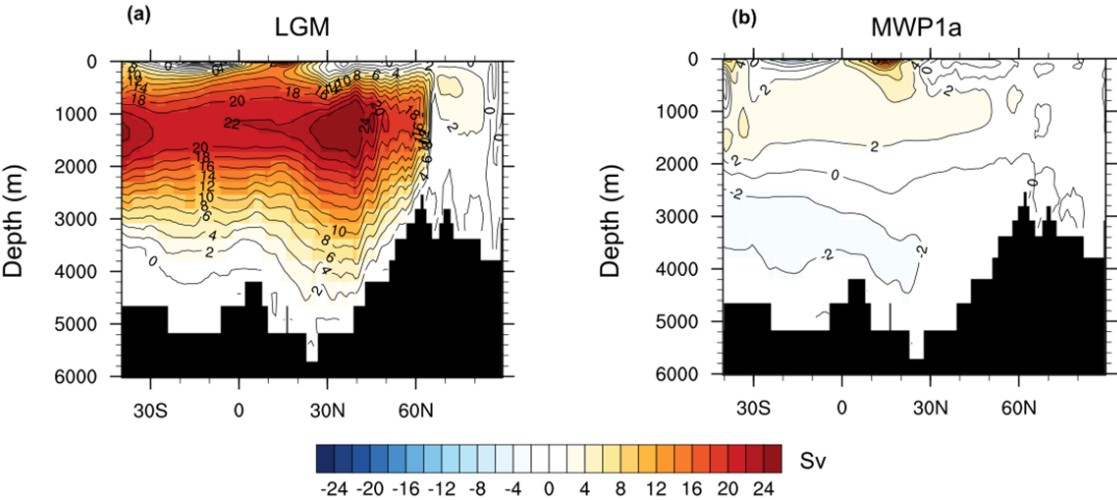

**Figure 3.** Cross-section of the AMOC streamfunction for the Last Glacial Maximum at 21 ka (a) and during MWP1a at 14.5 ka (b).

For the land part, the total carbon in all terrestrial carbon pools (litter, soil and vegetation) increases from 922.9 GtC to 1302.7 GtC between 21-15 ka and reaches 1563.6 GtC at 12 ka (Figure 2g). This increase is explained by the progressively warmer climate during the deglaciation and by the increase of $C_3$ plants and Gross Primary Productivity (GPP) on land. Even if our simulation currently doesn't continue beyond 12 ka, the total land carbon content evolution since the Last Glacial Maximum is within the range of other modelling studies like Prentice et al. (2011), O'ishi and Abe-Ouchi (2013), Ganopolski and Brovkin (2017), Jeltsch-Thömmes et al. (2019) and Müller and Joos (2020). Furthermore, in a setup with a fully coupled land and ocean carbon cycle it would be the ocean that supplies this additional land carbon, which would demand a constant outgassing over the period during which the land carbon inventory increases. During MWP1a, the $C_3$ cover fraction and the land GPP decrease, indicating the replacement of trees by grasslands and resulting in a decrease of 62.38 GtC stored in litter and vegetation pools as shown in Fig. 2g.

We can also evaluate the ability of the model to reproduce the biome distribution since the LGM in comparison to pollen data like the BIOME6000 Version 4.2 reconstruction (Harrison, 2017) based on the Palaeovegetation Mapping Project (Prentice and Webb III, 1998; Prentice et al., 2000; Harrison et al., 2001; Bigelow et al., 2003; Pickett et al., 2004). To do so we used the biomisation technique developed in Dallmeyer et al. (2019) to convert the different PFT cover fractions modelled in JSBACH into 9 biomes.We also used the best neighbour score (BNS) metric method presented in Dallmeyer et al. (2019) to quantify the similarity between the modelled biomes and the pollen data from the BIOME6000 database. This method uses the surrounding grid boxes of the studied grid cell (in each direction of the T31 grid) to compare with the pollen record. The agreement for each record is calculated with the distance weight of the best neighbour in each neighbourhood (using a Gaussian function) and varies between 1 if the modelled biomes in the grid box indicates the same biome as reconstructed and 0 if all grid cells in the neighbourhood disagree with the record. The BNS is the mean of all individual neighbourhood scores. The LGM modelled biomes on Fig. 4a show an overall good agreement with the pollen data with a total BNS value of 0.52. At high latitudes of the Northern Hemisphere, tundra and boreal forests are simulated in regions that are not covered by ice, which is consistent with the few pollen datasets available at these locations (BNS value of 0.78 and 0.19 respectively). Temperate forest is modelled over part of North America, grassland over Europe and temperate/warm forest over East Asia. This is in agreement with the pollen record even if some local discrepancies are observed like in central Asia. At low latitudes the model mostly reproduces the tropical forest (over Eastern South America, West Africa and Indonesia) as observed in the pollen data with a BNS value of 0.38 (Figure 4a).

Although the LGM conditions were different from those at 15 ka before MWP1a, in absence of other global reconstructions we also used the LGM BIOME6000 pollen record to compare to model results. According to our model, the biome distribution doesn't change much between 21 and 15 ka (Figure 4a,b) so that for many regions, the LGM pollen data show the same pattern as the simulated biomes at 15 ka. The BNS value at 15 ka is similar to the one at 21 ka for tropical forest, warm forest, savanna and desert. However, climatic differences between these two periods lead to small differences between the simulated biomes at 15 ka and the 21 ka pollen data which explains the lower total BNS value of 0.45 compared to 0.52 (Figure 4a,b). Part of the LGM tundra at high latitudes of the Northern Hemisphere is replaced by the boreal forest or grassland at 15 ka. At low

latitudes, there is a slightly larger extent of the temperate forest over East Asia and of the tropical forest over South America at 15 ka. The tropical forest over Indonesia is however already present since the LGM. We can focus on this region in more detail since most of the terrestrial organic matter inputs during MWP1a comes from this area (see next sections). Dubois et al. (2014) used isotopic composition of vascular plant fatty acid ($\delta^{13}C_{FA}$) from surface sediments in Indonesian seas to infer past regional vegetation over Indonesia during the LGM. They showed that during this time period the predominant vegetation was characterized by $C_3$ plants over central Indonesia. Indeed, even if the climate was colder and drier than during the Preindustrial, it was not sufficiently so to alter the vegetation distribution and decrease the rainforest coverage. Our biome reconstruction also agrees with modelling studies like the one of Cannon et al. (2009) or Prentice et al. (2011) that show using a dynamic vegetation model that tropical forest dominated the Sunda shelf during the Last Glacial Maximum. Recently, Dallmeyer et al. (2019) also evaluated the vegetation reconstruction from 4 different Earth System Models using the same harmonization method for PFT distribution that we used for Fig. 4. All ESMs used in Dallmeyer et al. (2019) model tropical forest over Indonesia during the LGM. All together, these results support the conclusion that tropical forests developed since the LGM continued to do so until at least 15 ka, even if the climate was colder and drier than the Preindustrial.

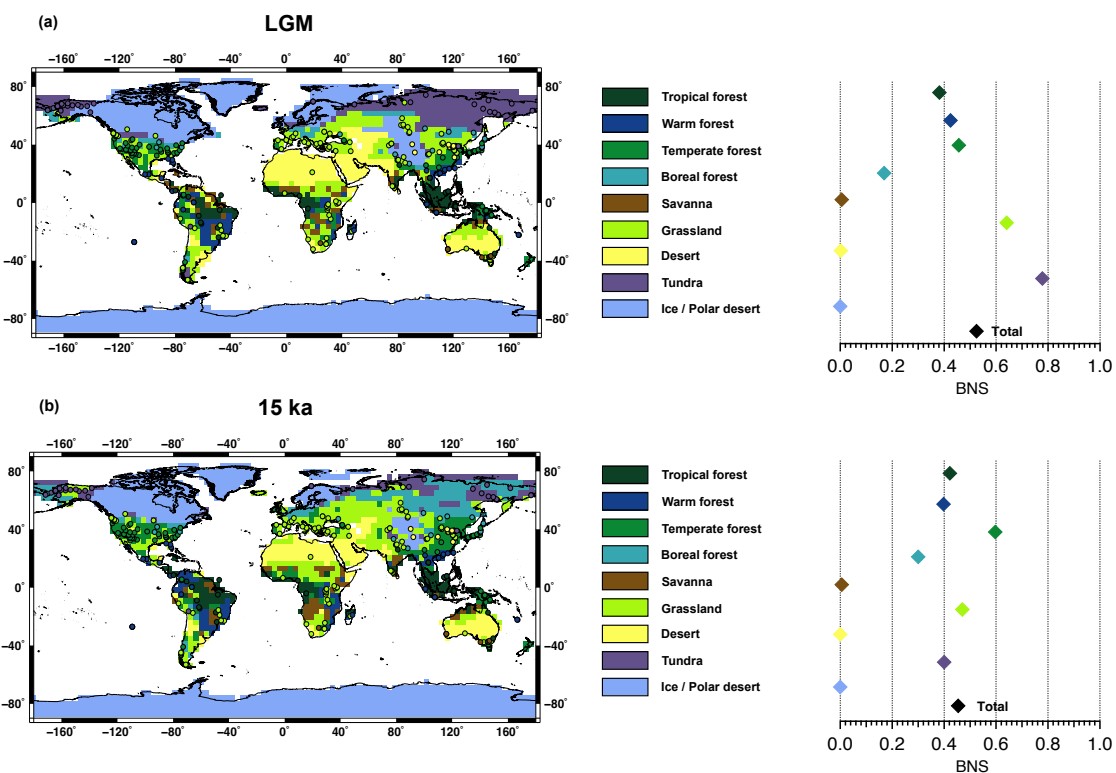

**Figure 4.** Biome distribution modelled by JSBACH at 21 ka (a) and 15 ka (b). The superimposed circles are the pollen data from the BIOME6000 Version 4.2 reconstruction at 21 ka for both figures (Harrison, 2017). The right plots indicate the best neighbour score, i.e. the similarity between the modelled biomes and the pollen data, for both time period.

## 3.2 Land-sea carbon fluxes

The transient adaptation of the land-sea mask and bathymetry during the deglaciation allows for flooded coastal land areas to be accounted for when the sea level changes. During the deglaciation, before and after Meltwater Pulse 1a (i.e. 21-15 ka and 14-12 ka), flooding is observed on almost all continental coasts but with a higher number of flooded land coastal regions located above 60° N (Figure 5). During MWP1a, local coastal land surfaces are flooded in East Asia, Indonesia and Australia, but again with a higher number of the flooded grid cells in northern Europe since the major source of the meltwater is in the

Northern Hemisphere (Figure 5). In terms of area, the flooded high latitudes coastal regions represent 1.03 x $10^6$ km$^2$, which is larger than the 8.04 x $10^5$ km$^2$ of flooded coastal area in Indonesia.

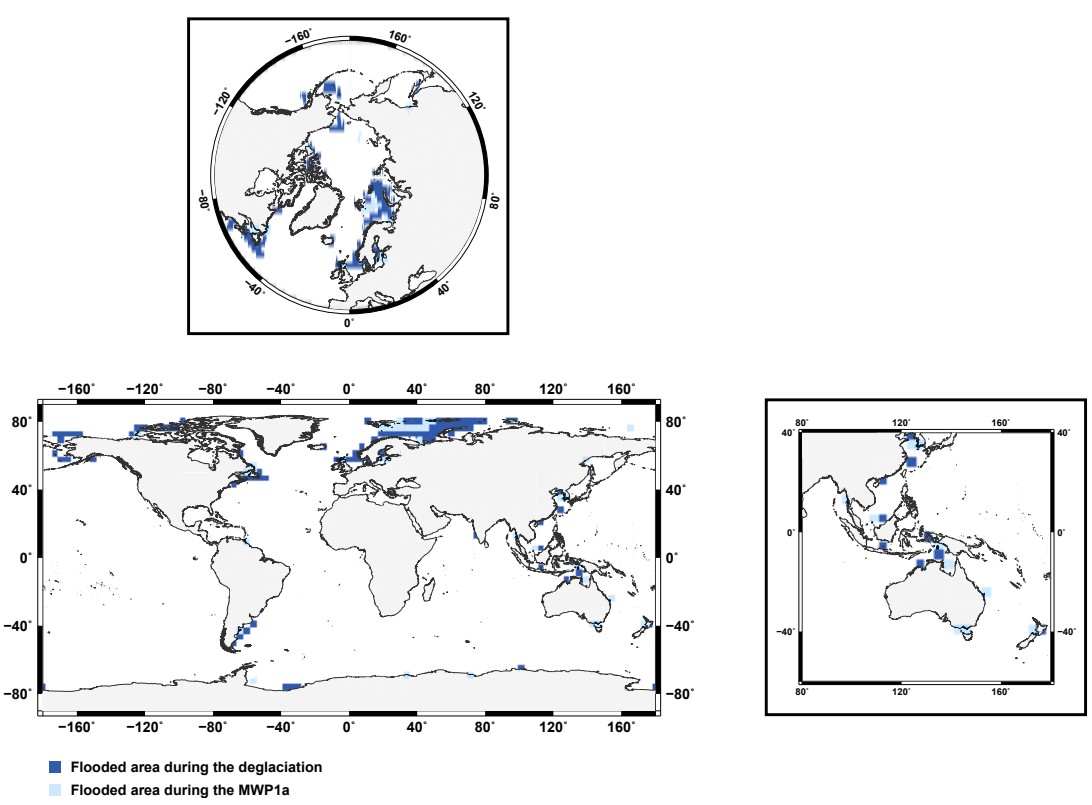

**Figure 5.** Spatial distribution of the coastal flooded areas during the last deglaciation (dark blue) and during the specific interval of MWP1a between 15-14 ka (light blue). Zooms over the Arctic and Indonesian regions are also shown.

Organic matter fluxes from land to ocean are computed during the different flooding events and the results obtained from the transient simulation show that terrestrial inputs are frequent over the last deglaciation but most of them happen after 15 ka (Figure 6). There are also differences in the size of the carbon fluxes depending on the origin of the terrestrial organic matter.

During MWP1a, 7.7 GtC are emitted to the atmosphere (related to the green vegetation), which corresponds to 26.6 % of

the total terrestrial organic matter inputs (Table 2). We however refrain from any discussion since these fluxes don't have an impact on the climate in this simulation with prescribed $CO_2$ concentrations. 21.2 GtC goes to the ocean, i.e. 73.4 % of the total terrestrial organic matter inputs (Table 2).

| | Green biomass and non-woody litter | Wood | Woody litter above ground | Woody litter below ground | Humus | Total |
|---|---|---|---|---|---|---|
| Amount in GtC between 21-12 ka | 25.1 (28.4 %) | 26.2 (29.7 %) | 7.3 (8.3 %) | 3.2 (3.6 %) | 26.5 (30 %) | 88.3 (100 %) |
| Amount in GtC between 15-14 ka | 7.7 (26.6 %) | 9.0 (31.1 %) | 2.6 (9 %) | 1.2 (4.2 %) | 8.4 (29.1 %) | 28.9 (100 %) |
| Destination | Atmosphere | Ocean-sediment interface | Water column | Sediment | Sediment | Atmosphere and ocean |

**Table 2.** Carbon mass of terrestrial organic matter going to the atmosphere, ocean-sediment interface, water column and sediment after the flooding of the coastal land areas during the last deglaciation and MWP1a.

Looking at the origin of the terrestrial organic matter inputs to the ocean, wood and humus are the largest contributors with
31.1 and 29.1 % of the total inputs respectively (atmosphere and ocean) and have even higher contribution of 42.5 and 39.6 % for the ocean part only. In comparison to the entire deglaciation between 21-12 ka, the flooding induced terrestrial carbon emissions and the terrestrial carbon inputs to the ocean represent a similar proportion than during MWP1a with respectively 28.4 and 71.6 % for a total of 88.3 GtC (Table 2). Wood and humus are again the major contributors of these land-sea fluxes. We have to emphasize that these numbers are for one millennium and thus the total terrestrial organic matter contribution to
the ocean of 21.2 GtC is non negligible compared to the entire deglaciation (63.2 GtC). However, in comparison to the ocean inventory of the Mixed Layer Depth of around 600 GtC or even of the global ocean of around 36000 GtC, we can consider that the terrestrial organic matter fluxes to the ocean are rather small (3.5 % of the Mixed Layer Depth inventory and 0.06 % of the global ocean inventory). Their effects on the ocean biogeochemistry are investigated next in Section 3.3.

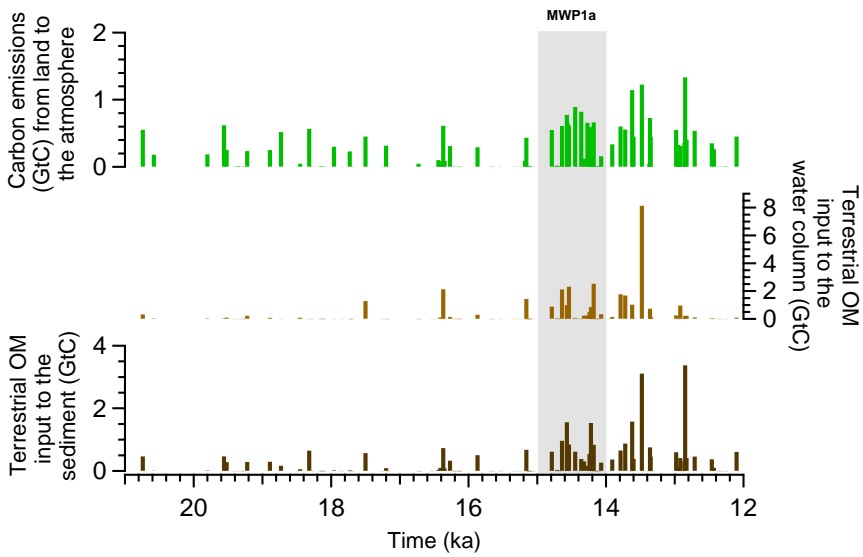

**Figure 6.** Time series of terrestrial organic matter inputs during the last deglaciation. Top panel: Flooding induced terrestrial carbon emissions (vegetation, green biomass and non-woody litter) to the atmosphere. Middle panel: Terrestrial organic matter (wood and woody litter above ground) inputs to the water column and to the water-sediment interface. Bottom panel: Terrestrial organic matter (woody litter below ground and humus) inputs to the sediment.

Among all contributors of terrestrial organic matter during MWP1a, the Indonesian region is the largest one with inputs to the
water column that represent 66.4 % of the total inputs (Figure 7a) and explained by wood inputs. The second largest contributor during this meltwater pulse is Australia with 17.3 %, followed by East Asia, the high latitudes of the Northern Hemisphere, South America and Europe with respectively 11.1, 2.2, 1.8 and 1.2 %. All these terrestrial inputs are dominated by wood inputs. As for the part that goes to the sediment, Indonesia and East Asia regions are also major contributors with 43.7 and 18.7 % of the total terrestrial inputs to the sediment during MWP1a (Figure 7b), mostly due to humus. Other regions subject to flooding
like Australia, the high latitudes of the Northern Hemisphere and Europe also contribute to terrestrial organic matter inputs to the sediment in smaller amounts. So even if most of the flooded areas are located in the Northern Hemisphere during MWP1a, their contributions of terrestrial inputs to the ocean are less significant than equatorial and low latitudes flooded areas.

These results are however only representative for a short time period relative to the rapid change induced by the meltwater
pulse and could be different from the rest of the deglaciation. To investigate this, we also present the contribution of these same areas during the full deglaciation (21-12 ka). In comparison to the specific case of MWP1a, Europe, South America, East Asia and the high latitudes of the Northern Hemisphere have similar contributions of terrestrial carbon and nutrients to the water column over the entire deglaciation (Figure 7a). In contrast, North America (East and West) and land areas in the high latitudes of the Southern Hemisphere show organic matter inputs of a few percent during the deglaciation, but not during
the large meltwater pulse. But the largest contribution of terrestrial organic matter to the water column observed during the

deglaciation, i.e. Indonesia, occurs during the large meltwater pulse. In terms of regional contribution to the total terrestrial inputs to the sediment, Indonesia is once again the largest contributor with 43.7 % (Figure 7b). East Asia contributes with 18.7 %, similar to Australia, but three times more important than Europe and the high latitudes of the Northern Hemisphere with 6.5 %. Among the smallest contributors are North and South America between 1 and 3 %. Once in the sediment, part of the terrestrial organic matter is buried, i.e. 0.042 and 0.168 GtC respectively for the woody litter and humus during the entire deglaciation. Inputs to the water column due to sediment erosion are of the order of 0.0026 and 0.026 GtC for the woody litter and humus respectively. The remaining part is remineralized.

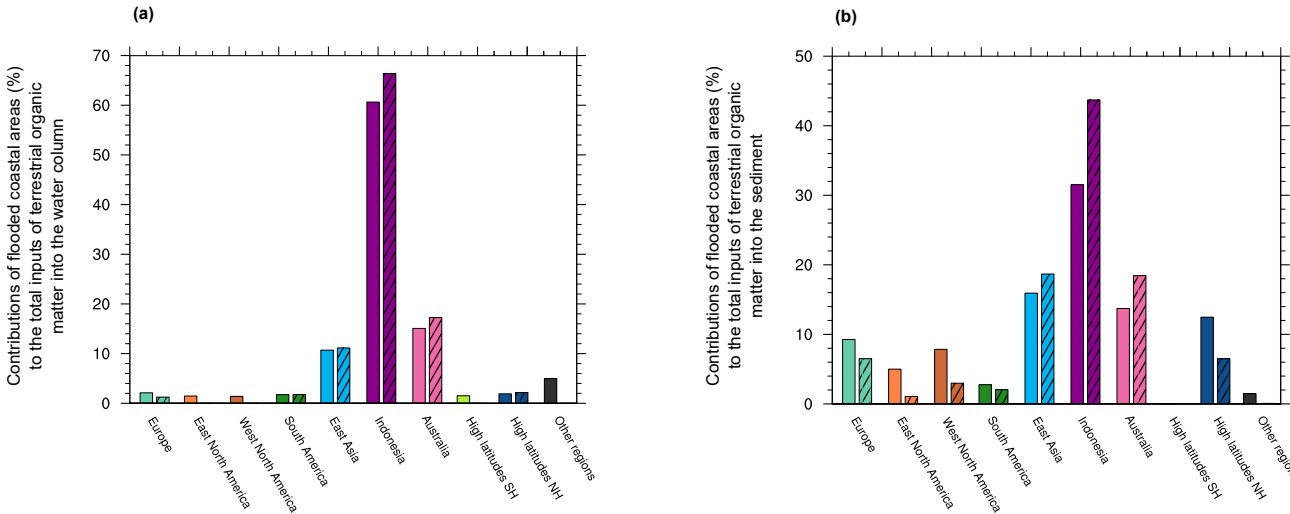

**Figure 7.** Contributions of flooded coastal areas to the total terrestrial organic matter inputs (considered as 100 %) that go to the water column (a) and to the sediment (b) during the full deglaciation between 21-12 ka (full bars on the left) and during MWP1a between 15-14 ka (hatched bars on the right).

## 3.3   Implications for the ocean biogeochemical cycle

To assess the role of the terrestrial organic matter inputs on ocean biogeochemistry during MWP1a, we performed a sensitivity experiment without terrestrial organic matter fluxes between land and ocean (as described in Section 2.4). The sensitivity experiment is branched off at 15 ka from the reference deglaciation run. Figure 8 shows anomalies of surface alkalinity, dissolved inorganic carbon and phosphate between simulations with and without terrestrial organic matter inputs (i.e. deglaciation reference run - sensitivity experiment) as average over MWP1a. An equatorial box between 15°N and 15°S around Indonesia is defined and subdivided in three parts: northern part, central part and southern part, to discuss the largest differences (see Figure 9).

The surface alkalinity anomaly shows regionally higher values of around 10 mmol m$^{-3}$ in central Indonesia, East Asia and up to 22 mmol m$^{-3}$ at high latitudes of the Northern Hemisphere where coastal areas are flooded (Figure 8a). The largest difference between the two simulations is observed between Indonesia and Australia with values from 148 to 214 mmol m$^{-3}$. In contrast, lower values are observed in the Atlantic Ocean, in the West Equatorial Pacific and in the Arctic Ocean. A similar pattern is observed for the dissolved inorganic carbon with higher values between 0 and 30 mmol C m$^{-3}$ in Indonesia, East Asia and high latitudes of the Northern Hemisphere (Figure 8b). Several grid points in the equatorial box show higher values between the two simulations with 36 mmol C m$^{-3}$ (northern part), 90 mmol C m$^{-3}$ (central part) and 228 mmol C m$^{-3}$ (southern part). These positive anomalies (detailed in Fig. 9a for alkalinity and Fig. 9b for DIC) are associated with the input of terrestrial organic matter, mainly wood and humus, during a flooding event with additional carbon and nutrients entering the ocean leading to an increase of the surface alkalinity and DIC once the terrestrial organic matter has been remineralized. At depth, an increase in DIC is observed below 1500 m in the Atlantic Ocean, Nordic Seas, Australia-Indonesia Coastal Province and Sunda-Arafura Shelves Province. For the surface phosphate anomaly, small differences are observed in the Atlantic part of the Southern Ocean and in the equatorial band with lower values up to 0.02 mmol P m$^{-3}$ (Figure 8c). Higher values up to 0.05 mmol P m$^{-3}$ are observed in West Indonesia, in the North Atlantic Ocean and at high latitudes of the Northern Hemisphere.

The temporal evolution of the ocean biogeochemistry is also quite similar between the simulations with and without terrestrial organic matter inputs to the ocean. The global surface alkalinity shows similar variations with values ranging from 2247 to 2282 mmol m$^{-3}$ between 15-14 ka (Figure 8d), mainly controlled by changes in the physical state of the ocean, i.e. sea surface salinity and temperature, in response to freshwater inputs and circulation changes. The largest difference observed between the two simulations occurs between 14.69-14.62 ka with the simulation with terrestrial organic matter fluxes being around 4 mmol m$^{-3}$ higher than the simulation without these fluxes. This is explained by land-sea fluxes with a total of 3.08 GtC. The global surface DIC and phosphate variations during MWP1a show a similar trend within the two simulations with values ranging respectively from 1916 to 1950 mmol C m$^{-3}$ (Figure 8e) and from 0.44 to 0.54 mmol P m$^{-3}$ (Figure 8f). The differences in the biogeochemistry within the two simulations highlight the fact that terrestrial organic matter inputs to the ocean, related to several flooding events during MWP1a, only have a relatively small effect of 1-2 % on the global surface alkalinity, dissolved inorganic carbon and phosphate. The land-sea fluxes are indeed relatively small (3.5 %) in comparison to the Mixed Layer Depth carbon inventory (upper 200 m of the water column).

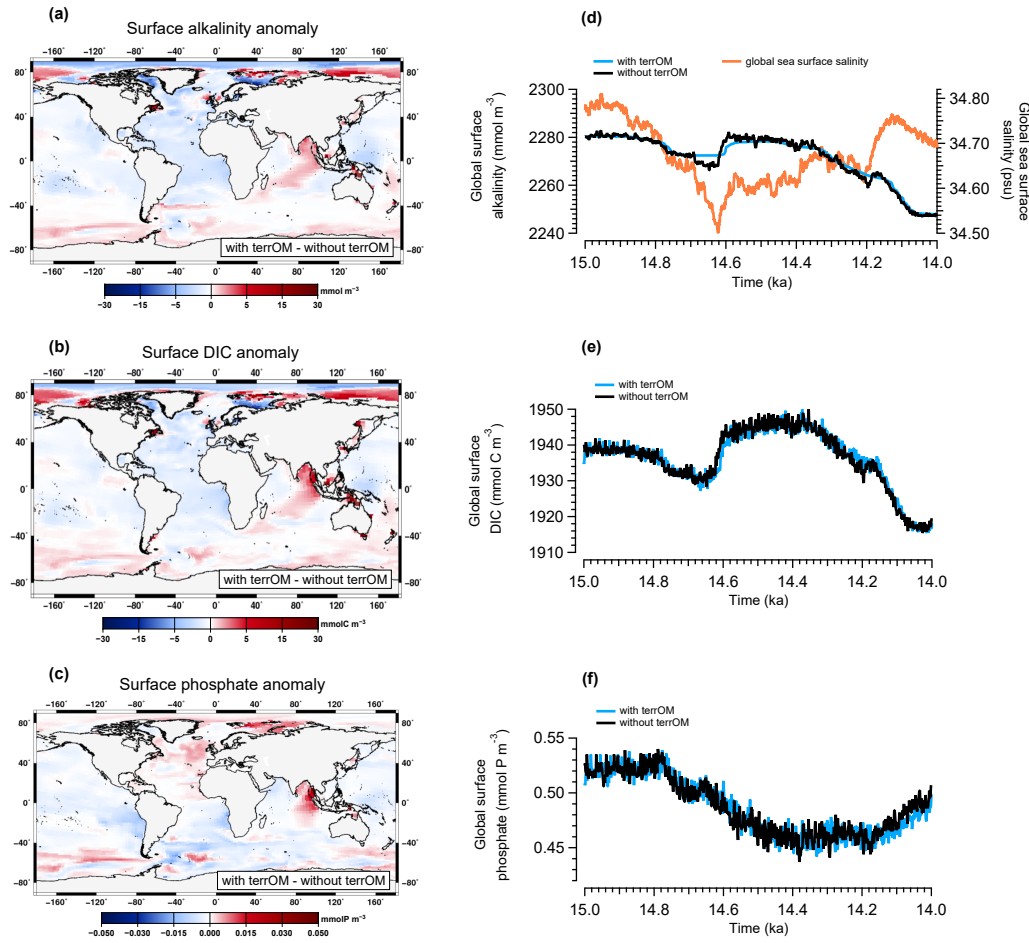

**Figure 8.** Anomaly of the mean surface alkalinity (a), surface dissolved inorganic carbon (b), and surface phosphate (c) between the simulations with and without terrestrial organic matter fluxes averaged over MWP1a. Time evolution of the two simulations during MWP1a for annual mean global surface alkalinity (d), surface DIC (e) and surface phosphate (f). The global sea surface salinity is also represented in orange on panel (d).

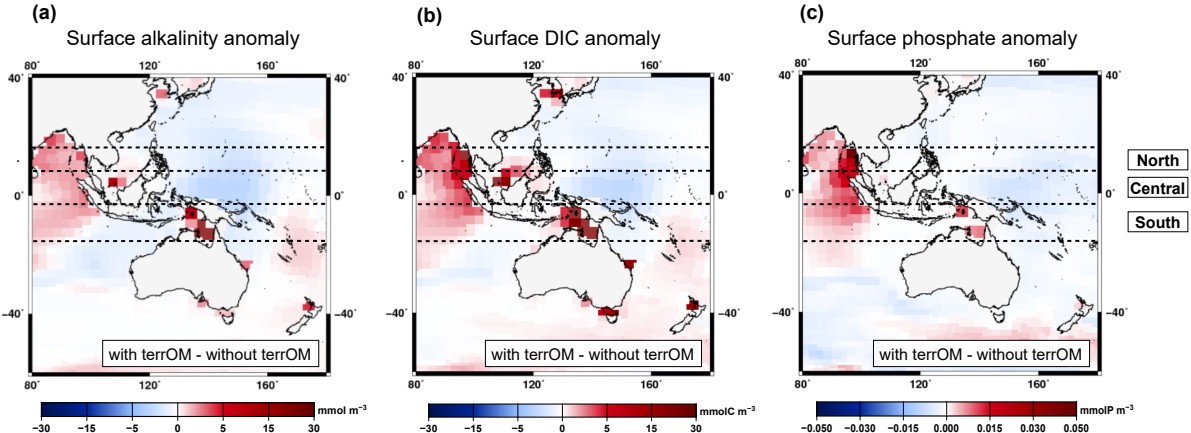

**Figure 9.** Same as Fig. 8a, b, c but with a focus over Indonesia for the surface alkalinity (a), surface dissolved inorganic carbon (b) and surface phosphate (c) between the two simulations with and without terrestrial organic matter fluxes averaged over MWP1a. An equatorial box is defined between 15°N and 15°S and subdivided in three areas: northern part, central part and southern part.

Due to the stoichiometry of the terrestrial organic matter being higher than of the marine organic matter (C:N:P = 3650:11:1 for the wood, 7600:51:1 for the woody litter and 465:10:1 for the humus in comparison to 122:16:1 for marine organic mat-
ter) there is an excess of carbon in the ocean once the terrestrial organic matter has been remineralized. Indeed, net primary production uptakes only 122 moles of carbon per mole of phosphate which leaves a large fraction of remineralized terrestrial carbon in the water column. This leads the ocean to outgas $CO_2$ to the atmosphere. Similarly to what has been presented above, we investigate the effect of terrestrial organic matter inputs on the $CO_2$ fluxes between ocean and atmosphere during MWP1a. Figure 10 shows the anomaly between the two simulations (with and without land-sea fluxes) of the mean surface $CO_2$ flux
between the ocean and atmosphere. Indonesia shows differences with higher $CO_2$ outgassing values of up to $2.2 \times 10^{-9}$ kg C $m^{-2}$ $s^{-1}$ for the simulation taking into account the land-sea fluxes. These larger values originate from the remineralization of the wood and humus associated with tropical forest developed at that time, as explained in Section 3.1 and shown in Fig. 4. Other flooded areas during MWP1a also contribute to higher $CO_2$ fluxes to the atmosphere for the simulation with terrestrial organic matter fluxes, like in East Asia with $0.17\text{-}0.22 \times 10^{-9}$ kg C $m^{-2}$ $s^{-1}$ or in the northern Australian coast with $0.36\text{-}0.39$
$\times 10^{-9}$ kg C $m^{-2}$ $s^{-1}$. Overall, the terrestrial organic matter fluxes only have a local influence on the ocean behaviour with higher outgassing in the Indonesian region during the millennial event of MWP1a.

To investigate the influence of the carbon to nutrients ratio of terrestrial organic matter on the $CO_2$ fluxes between the ocean and atmosphere during MWP1a we performed two additional sensitivity experiments following the procedure described in Section
2.4. The first sensitivity experiment uses a higher C:N:P ratio of terrestrial organic matter, arbitrarily defined as twice the carbon and nitrogen values of Goll et al. (2012) used in the reference simulation since no higher estimations exist in the literature. The second sensitivity experiment uses a lower C:N:P ratio of terrestrial organic matter based on studies from Lerman et al.

(2004) and Cleveland and Liptzin (2007). These changes in the stoichiometry also require an adjustment of the $O_2$ and $NO_3^-$ demand and corresponding $\Delta Alk$ changes during remineralization of terrestrial organic matter. All values are summarized in Table 3.

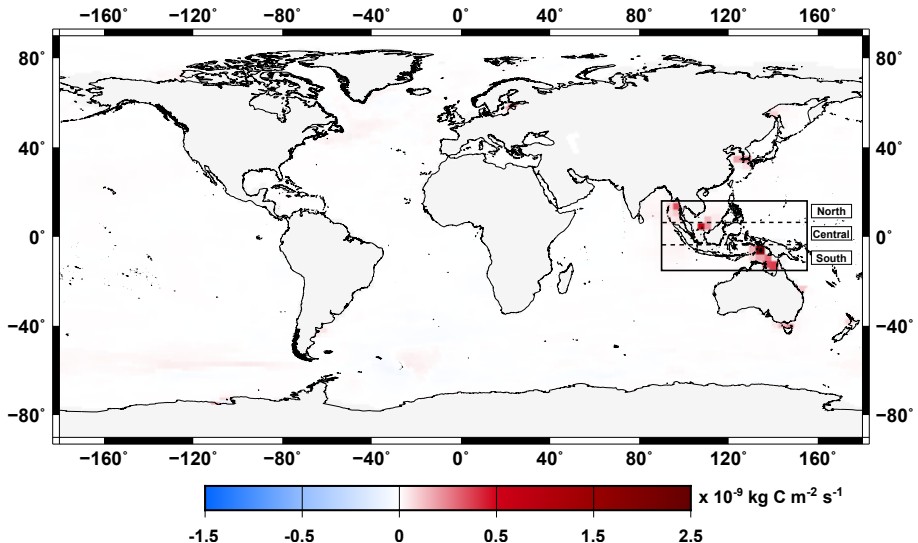

**Figure 10.** Anomaly of the mean surface $CO_2$ flux between the simulations with and without terrestrial organic matter inputs to the ocean averaged over MWP1a. Negative values indicate flux from the atmosphere to the ocean (uptaking) and positive values indicate flux from the ocean to the atmosphere (outgassing). The equatorial box between 15°N and 15°S highlights locations with the largest differences.

|  |  |  | Aerobic remineralization | | Anaerobic remineralization | |
|---|---|---|---|---|---|---|
|  | Terrestrial OM | C:N:P | $O_2$ demand | $\Delta Alk$ | $NO_3^-$ demand | $\Delta Alk$ |
| High stoichiometry | Wood | 7300:22:1 | 7344 | -23 | 5853.2 | 5852.2 |
| sensitivity test | Woody litter | 15200:102:1 | 15404 | -103 | 12221.2 | 12220.2 |
|  | Humus | 930:20:1 | 970 | -21 | 756 | 755 |
| Low stoichiometry | Wood | 510:4:1 | 518 | -5 | 410.4 | 409.4 |
| sensitivity test | Woody litter | 186:13:1 | 212 | -14 | 156.6 | 155.6 |
|  | Humus | 140:6.6:1 | 153.2 | -7.6 | 115.96 | 114.96 |

**Table 3.** Terrestrial organic matter stoichiometry with consumption of oxygen, nitrate and change in alkalinity during remineralization for the sensitivity experiments with high and low stoichiometry. For comparison, values for the reference deglaciation simulation are given in Table 1.

We compare the largest outgassing events over Indonesia for the different simulations with and without terrestrial organic matter as well as for the simulations with different carbon to nutrients ratios for this organic matter. As already observed in Fig. 10, the outgassing of the ocean in the equatorial region of Indonesia during MWP1a is driven by the terrestrial organic matter fluxes during flooding events. We observe a $CO_2$ flux to the atmosphere happening concurrently to the flooding events and then

slowly decreasing with time until all the terrestrial organic matter has been remineralized. In northern part (12.5° N - 96.5° E), the outgassing starts at 14.64 ka, reaches 3.8 x $10^{-9}$ kg C m$^{-2}$ s$^{-1}$ and then decreases for 200 years which corresponds to the decay time of the wood material (i.e. stems) in the ocean (Figure 11a). This outgassing event is due to the transfer during the flooding event of terrestrial material to the ocean, dominated by the wood with 1.72 GtC and by the humus with 0.79 GtC. For the simulation without terrestrial organic matter inputs, the ocean behaves differently and rather uptakes carbon with a slightly

negative $CO_2$ flux. This opposite behaviour between the two simulations with and without terrestrial organic matter fluxes is also observed in central and southern Indonesia and highlights the key role of terrestrial organic matter fluxes in the oceanic outgassing. In central part (4.5° N - 107.5° E) we observe an outgassing peak of 8.8 x $10^{-9}$ kg C m$^{-2}$ s$^{-1}$ at 14.54 ka and then a positive $CO_2$ flux for 300 years (Figure 11b). This outgassing event is primarily the consequence of wood input to the ocean with 1.95 GtC, as well as humus input, the second largest contributor, with 0.69 GtC. In southern part (6.5° S - 134.5° E) we

also have an outgassing with 5.3 x $10^{-9}$ kg C m$^{-2}$ s$^{-1}$ to the atmosphere at 14.18 ka due to dominant wood inputs of 2.15 GtC (humus inputs represent 0.67 GtC) (Figure 11c).

For a higher carbon to nutrients ratio the three locations in Indonesia reproduce similar $CO_2$ fluxes to the atmosphere than those in the reference simulation. For a lower carbon to nutrients ratio, the $CO_2$ flux is smaller than the reference simulation with only slightly positive values following the flooding event, with maximum of 0.085 x $10^{-9}$ kg C m$^{-2}$ s$^{-1}$ for northern

part, 0.67 x $10^{-9}$ kg C m$^{-2}$ s$^{-1}$ for central part and 0.14 x $10^{-9}$ kg C m$^{-2}$ s$^{-1}$ for southern part (Figure 11). These sensitivity experiments highlight the fact that even with a very high carbon to nutrients ratio (so larger fraction of remineralized terrestrial carbon in the water column), the outgassing $CO_2$ flux doesn't increase. However, in the case of the lower carbon to nutrients ratio for terrestrial organic matter, the $CO_2$ flux to the atmosphere is greatly reduced. But again, these fluxes are only happening at regional hotspots and do not affect the global net $CO_2$ flux between the ocean and the atmosphere.


Besides the C:N:P ratios, the remineralization rates of the terrestrial organic matter in sea water are not well constrained parameters. The choice of different rates could lead to higher or lower $CO_2$ flux to the atmosphere. In the deglaciation run and presented sensitivity simulations with higher and lower stoichiometries of terrestrial organic matter, the remineralization rates were prescribed to 2.7 x $10^{-5}$ d$^{-1}$ for wood, 2.7 x $10^{-4}$ d$^{-1}$ for woody litter and 5.5 x $10^{-4}$ d$^{-1}$ for humus. The new values

in this sensitivity experiment are 1.0 x $10^{-4}$ d$^{-1}$ for wood, 2.0 x $10^{-3}$ d$^{-1}$ for woody litter and 8.0 x $10^{-3}$ d$^{-1}$ for humus. This simulation uses the same higher stoichiometry ratios as one of the the first sensitivity studies (see Table 3) to get an upper estimate of the potential impact of terrestrial fluxes.

We observe higher $CO_2$ outgassing in the defined equatorial box over a shorter time period (Figure 11). For the northern part, the $CO_2$ flux to the atmosphere reaches 20 x $10^{-9}$ kg C m$^{-2}$ s$^{-1}$ after the flooding at 14.64 ka and decreases twice as fast as

the simulation with high stoichiometry (Figure 11a). Similar behaviour is observed for central and southern part with an out-

gassing peak after the flooding at 14.54 ka and 14.18 ka of respectively $32 \times 10^{-9}$ kg C m$^{-2}$ s$^{-1}$ and $27 \times 10^{-9}$ kg C m$^{-2}$ s$^{-1}$ (Figure 11b,c). This increased $CO_2$ flux to the atmosphere is primarily a result of wood remineralization since, as for previous simulations, wood dominates the terrestrial organic matter input to the ocean during flooding events at that latitude. Since wood is not buried in the sediment, the amount of material that can be remineralized is the same as in previous simulation, but at
faster rate. Part of the outgassing is still due to the remineralization of woody litter and humus before they are buried.

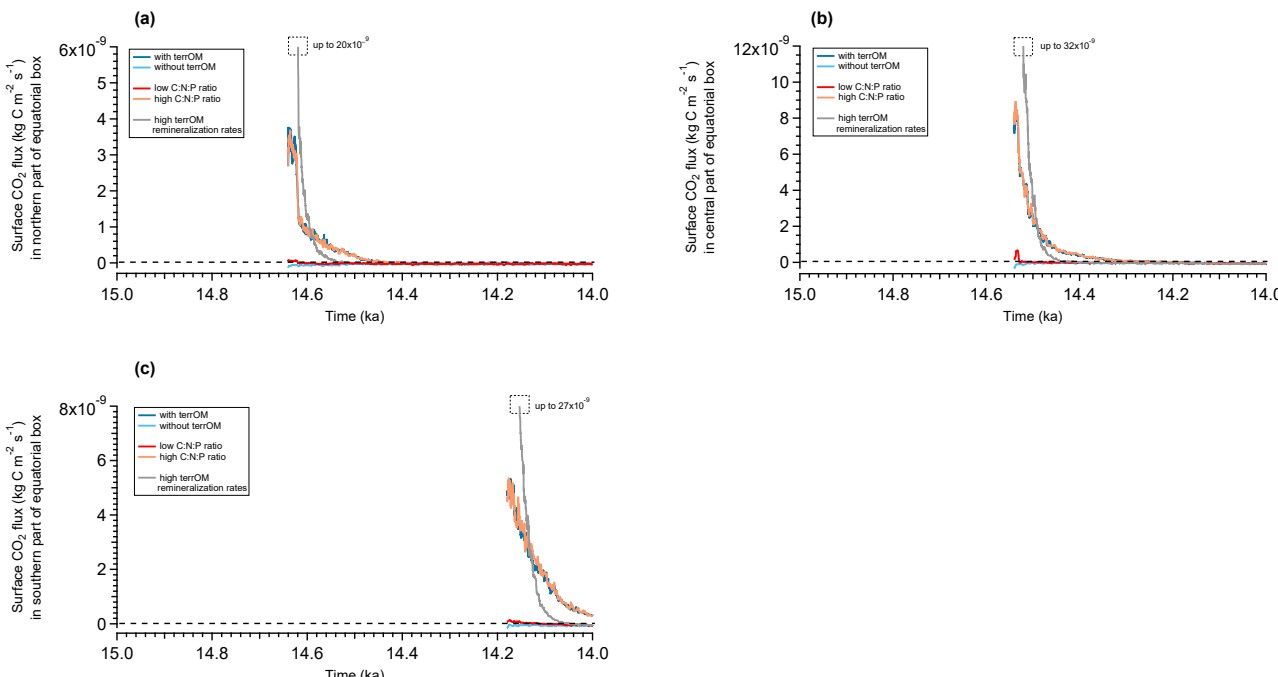

**Figure 11.** Evolution of the surface $CO_2$ flux during MWP1a for flooded grid cells in northern part (a), central part (b) and southern part (c) of the equatorial box defined between 15°N and 15°S for 5 different simulations: the reference simulation with the terrestrial organic matter fluxes (dark blue), the sensitivity experiment without terrestrial organic matter fluxes (light blue), the sensitivity experiment with low stoichiometry for terrestrial organic matter (red), the sensitivity experiment with high stoichiometry (orange) and the sensitivity experiment with high stoichiometry and high remineralization rates of terrestrial organic matter (grey). 50 years running mean are plotted for each simulation. Positive values indicate an outgassing to the atmosphere and negative values indicate an uptaking by the ocean. The time series start when land is flooded.

## 4   Conclusions

In this study we present for the first time the implementation of terrestrial organic matter fluxes between land and ocean at transiently changing land-sea interface in the global Earth System Model MPI-ESM. This unique setup of MPI-ESM was used to perform a transient deglaciation simulation from 21 to 12 ka accounting for sea level rise induced by meltwater inputs

from ice sheets and consequential changes in ocean depth and coast lines. The period between 15-14 ka, which corresponds to Meltwater Pulse 1a, has been highlighted because it is characterized by larger terrestrial organic matter inputs to the ocean than during the first part of the deglaciation. Indeed, a total of 21.2 GtC, mostly arising from Indonesia, goes to the ocean during this millennial event, which represents 34 % of the total amount of terrestrial organic matter entering the ocean over the last deglaciation. This terrestrial carbon is remineralized once in the ocean within a time frame of hundreds of years. The effect of

this supplementary carbon brought from land is only observed at regional hotspots with local outgassing to the atmosphere. The carbon input doesn't seem to be large enough to impact the global behaviour of the ocean, considering that it represents a very small amount in comparison to the global ocean inventory (0.06 %). A sensitivity experiment also emphasizes that the terrestrial organic matter fluxes only have a small effect on the surface alkalinity and dissolved inorganic carbon (around 1-2 % increase). However, the local $CO_2$ fluxes between the ocean and the atmosphere during MWP1a are driven by the terrestrial

organic matter inputs. This regional outgassing to the atmosphere observed in Indonesia is explained by wood inputs and is supported by several lines of evidences suggesting that prior to this meltwater pulse event and even as far back as the Last Glacial Maximum, tropical forest was developed in this region favouring the storage of carbon-rich material on land entering the ocean once the land is flooded. As a complement, an additional set of sensitivity experiments show that the magnitude of outgassing during MWP1a is insensitive to higher carbon to nutrients ratio of the terrestrial organic matter but rather responds

to higher remineralization rates of terrestrial organic matter. Overall, our simulation is a first step towards a fully coupled ESM including carbon and nutrients fluxes at the land-sea continuum that will be applicable for long transient paleo-simulations over the last glacial/interglacial cycles.

*Author contributions.* TE designed the study and wrote the manuscript. KDS and HP realized the model development. KDS conducted the deglaciation simulation and TE performed the sensitivity experiments. All co-authors contributed to the manuscript with valuable comments

and discussions.

*Competing interests.* The authors declare that they have no conflict of interest.

*Acknowledgements.* We acknowledge support through the project PalMod, funded by the German Federal Ministry of Education and Research (BMBF). Simulations were performed at the German Climate Computing Center (DKRZ). We thank Thomas Riddick for internal review of this manuscript. We also thank Anne Dallmeyer for her help with the biome reconstruction.

*Financial support.* This research has been supported by the German Federal Ministry of Education and Research (PalMod initiative, FKZ: grant no. 01LP1925C). The article processing charges for this publication are covered by the Max Planck Society.

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
