# Peer review of "Local oceanic CO2 outgassing triggered by terrestrial carbon fluxes during deglacial flooding"

_Climate of the Past, 2021_

## Author Comment (AC2)

We thank Reviewer 1 for the detailed review which helped to improve the manuscript. We are providing our answers (in blue) to the comments and will revise the manuscript accordingly.

1) Title: I suggest "Local oceanic CO2 outgassing…."

We agree that this suggested title is indeed more appropriate and highlights the more local impact of the investigated processes. We changed the title of the manuscript to "Local oceanic $CO_2$ outgassing triggered by terrestrial carbon fluxes during deglacial flooding".

2) Introduction:

The structure of the introduction needs to be improved and some aspects that are the focus of the manuscript are currently missing. The introduction does not properly mention the current hypotheses to explain glacial/interglacial changes in $pCO_2$. L52-53 is confusing, as there has been a lot of work done in understanding glacial-interglacial changes in atmospheric $CO_2$, including some transient simulations. On the other hand, the introduction includes one paragraph on the impact of Heinrich events on the carbon cycle from a modelling perspective, but links with the current studies are not made. The introduction should also include a paragraph on estimates of glacial-interglacial changes in terrestrial carbon to provide a perspective on the modelling outputs. Finally, since MWP1A is mentioned throughout the manuscript, a brief paragraph on MWP1a should be added. This paragraph could describe estimates of the timing of MWP1A, its magnitude and the potential origin of this meltwater pulse (NH vs SH).

Thank you for pointing out these aspects. We revised the introduction with a more detailed section on glacial/interglacial $CO_2$ variations and on the processes of the ocean physics and biogeochemistry that can explain these $CO_2$ variations. We also added a new section focusing on MWP1a. The text of the introduction has been modified in the revised manuscript.

3) Deglacial sea-level rise

Since the results of the present study are dependent on the sea-level rise, a timeseries of the simulated deglacial sea-level rise along with paleo-estimates should be included in Figure 2. The implications of potential differences between simulated and estimated deglacial sea-level rise should be discussed.

We added two curves in Fig. 2 (see panel b in the new figure below). The first one is the global sea level stack from Spratt and Lisiecki (2016) based on isotopic measurements. The second one is the modelled global sea level change based on the freshwater inputs to the ocean that are derived accordingly to the ice sheet volume change from the GLAC1D ice sheet reconstruction from Tarasov et al. (2012).

The global sea level change in the model increases by 67.4 m between 21-12 ka, which is close to the value of 69 m obtained from proxy data (Spratt and Lisiecki, 2016). During MWP1a, the global sea level changes in the model show some quantitative differences compared to Spratt and Lisiecki (2016) record. There are two main causes. First, uncertainties exist in the prescribed ice sheet reconstructions. For instance, the ice sheet volume and the timing of freshwater input show noticeable differences between GLAC1D and ICE6G reconstructions (see Ivanovic et al. 2016 for a comparison). Second, all the freshwater input to the ocean is treated as liquid water. The global sea level increases in the model of 19.6 m for the 500 years of largest freshwater inputs, which is in the high range of the previous estimations with a global

sea level increase from 8.6 to 20.2 m (Deschamps et al., 2012; Liu et al., 2016; Lin et al., 2021). Then, between 14-12 ka, the sea level in the model only slightly increases in comparison to the Spratt and Lisiecki (2016) record. We added this discussion in the main text of the revised manuscript.

[Figure]

[Figure]

Figure 2. Time series of land, ocean and atmosphere variables over the last deglaciation. The presented outputs start at 21 ka. (a) Freshwater input to the global ocean. (b) Global sea level estimate from Spratt and Lisiecki (2016) (light purple) and modelled in MPI-ESM based on the freshwater inputs (dark purple). (c) Atlantic Meridional Overturning Circulation streamfunction. (d) $CO_2$ concentration measured in ice cores (Köhler et al., 2017). (e) Modelled global net $CO_2$ flux between the ocean and the atmosphere. Positive $CO_2$ flux mean that the ocean is outgassing to the atmosphere and negative $CO_2$ flux mean that the ocean is uptaking carbon. (f) Global ocean net primary production. (g) Total carbon in all terrestrial carbon pools, i.e. vegetation, soil and litter. The thick darker curves are 500 years running mean for the panel (a) and 50 years running mean for the panels (c), (e) and (f). A zoom over MWP1a is presented on the right.

4) Line by line comments

P1, L. 1: The first sentence of the abstract is odd. It needs to be rephrased, and most likely split in two sentences.

We rephrased the first sentence to: "Exchanges of carbon between the ocean and the atmosphere are key processes that influence past climates via glacial/interglacial variations of the $CO_2$ concentration".

P1, L.2: I suggest "induces a sea-level rise"

Change done.

P1, L. 10: I suggest "leads to 21.2 GtC transfer of terrestrial organic carbon to the ocean"

Done.

P1, L. 20: "including" instead of "triggered"

Done.

P2, L. 28: Consider adding references to Lambeck et al. 2014.

We added this reference to the main text.

P7, L. 190: Weren't the nutrient concentrations adjusted for the lower sea-level at the LGM?

We initialized the model with nutrient concentrations from a present day MPI-ESM simulation. We didn't adjust for the 3.5 % change in oceanic volume between the present day and the LGM, i.e. the prescribed total nutrient inventory at LGM is slightly smaller than for Preindustrial. We specify this aspect in the main text of the manuscript.

P8, L. 218-220: I find this sentence confusing. What do you suggest the relationship between oceanic circulation and $CO_2$ uptake is? A well ventilated Southern Ocean is usually associated with $CO_2$ outgassing and not uptake.

We agree that this sentence is confusing so we removed it from the text. In general, MPI-ESM tends to have a higher anthropogenic $CO_2$ uptake in the Southern Ocean compared to other CMIP models (Nevison et al., 2016) linked to an interplay of ventilation, biological production and deep/intermediate water formation.

In addition, the ocean to atm. $CO_2$ flux shown in Fig. 2c is not really explained, and barely mentioned in the text. However, given the experimental setup, it might be simply responding to the forced changes in atmospheric $CO_2$, and to changes in surface solubility.

Yes, you are correct. We added the following text in the revised manuscript: "As we prescribe the atmospheric $CO_2$ concentration, we omit any interaction between the land and the marine carbon cycle. Thus, the air-sea exchange is primarily following the atmospheric $CO_2$ increase, modified locally by physically induced changes of the circulation, biogeochemistry and surface solubility."

A timeseries of atmospheric $CO_2$ should be included in figure 2 to better understand the ocean-atm $CO_2$ flux (2c), and maybe a timeseries of globally averaged SST.

Following this comment, we added the atmospheric $CO_2$ concentration measured in ice cores (Köhler et al., 2017) and prescribed in the model on the panel (d) in the new Fig. 2 (see figure above).

P9, L. 230: The Pa/Th record from the Bermuda rise suggests an AMOC weakening during HS1, but not necessarily during MWP1a. I think that the most recent chronology suggest the end of HS1 and thus beginning of Bolling Allerod at 14.6 ka, contemporary with the beginning of MWP1a.

Yes, this is correct, the Pa/Th maximum is observed during the HS1, so before 15 ka. We remove this sentence in the revised manuscript.

P12, L. 283 and throughout: I understand why you are referring to "terrestrial organic carbon input to the atmosphere", however this is not correct and could be confusing. It might be better to simply refer to "terrestrial carbon input to the atmosphere".

Thank you for addressing this issue on the terminology, which is indeed confusing. The correct wording must be emission of carbon origin from remineralization of short-living terrestrial organic matter. We replace "terrestrial organic carbon input to the atmosphere" in the manuscript by "flooding induced terrestrial carbon emissions".

P19, L. 369 and throughout manuscript: I am not sure that the use of North/central and south Indonesia is correct. Maybe it is more appropriate to refer to the Sunda and Sahul shelves.

We refrain here from using the terms "Sunda and Sahul shelves" or the name of islands (e.g. Borneo, Sumatra…) because the resolution of the model is too coarse to simulate the shelves or the individual small islands in Indonesia. Instead we define an equatorial box between 15°N and 15°S around Indonesia where the largest differences are observed, and divide this box in three parts: northern part, central part and southern part (see new Fig. 10). These three parts are then used for the discussion in the main text. Figure 9 has also been revised accordingly.

[Figure]

Figure 10. Anomaly of the mean surface $CO_2$ flux between the simulations with and without terrestrial organic matter inputs to the ocean averaged over MWP1a. Negative values indicate flux from the atmosphere to the ocean (uptaking) and positive values indicate flux from the ocean to the atmosphere (outgassing). An equatorial box is defined between 15°N and 15°S and subdivided in three areas: northern part, central part and southern part.

Figure 4: Comparing 15 ka vegetation with reconstructions from 21 ka does not seem appropriate. Please show the JSBACH field at 21 ka compared to LGM proxies. I however understand that given that the type of vegetation at the area of flooding will impact the terrestrial organic transfer, it might also be necessary to show JSBACH at 15ka.

Following your comment and a similar comment made by Reviewer 2, we added a new Fig. 4 to show the comparison between the modelled biomes and pollen data at 21 ka. We still show the comparison at 15 ka for the purpose of the manuscript since the discussion is focused on MWP1a.

[Figure]

Figure 4. Biome distribution modelled by JSBACH at 21 ka (a) and 15 ka (b). The superimposed circles are the pollen data from the BIOME6000 Version 4.2 reconstruction at 21 ka for both figures (Harrison et al., 2017).

We added the following text in the revised manuscript: "The LGM modelled biomes on Fig. 4a show an overall good agreement with the pollen data. At high latitudes of the Northern Hemisphere, tundra and boreal forests are simulated in regions that are not covered by ice, which is consistent with the few pollen datasets available at these locations. Temperate forest is modelled over part of North America, grassland over Europe and temperate/warm forest over East Asia. This is generally in agreement with the pollen record even if some local discrepancies are observed like in central Asia. At low latitudes the model mostly reproduces the tropical forest (over Eastern South America, West Africa and Indonesia) as observed in the pollen data (Figure 4a). Although the LGM conditions were different from those at 15 ka before MWP1a, in absence of other reconstructions we also used the LGM BIOME6000 pollen record to compare to model results. According to our model, the biome distribution doesn't change much between 21 and 15 ka (Figure 4a, b) so that for many regions, the LGM pollen data show the same pattern as the simulated biomes at 15 ka. However, climatic differences between these two periods lead to small differences between the simulated biomes at 15 ka and the 21 ka pollen data. Part of the LGM tundra at high latitudes of the Northern Hemisphere is replaced by the boreal forest or grassland at 15 ka. At low latitudes, there is a slightly larger extent of the temperate forest over East Asia and of the tropical forest over South America at 15 ka. The tropical forest over Indonesia is however already present since the LGM."

Figure 8: Add AMOC and/or meltwater timeseries?

We prefer not to add the AMOC or meltwater timeseries on the Fig. 8 since they are already shown in previous figures. Instead, we added on the panel (d) in the new Fig. 8 the global sea surface salinity evolution during MWP1a since changes in surface alkalinity are mainly controlled by changes in sea surface salinity, which are induced by freshwater inputs at that time. We also removed the indication of the timing of flooding events on the right figures and only mention them in the main text.

[Figure]

Figure 8. Anomaly of the mean surface alkalinity (a), surface dissolved inorganic carbon (b), and surface phosphate (c) between the simulations with and without terrestrial organic matter fluxes averaged over MWP1a. Time evolution of the two simulations during MWP1a for annual mean global surface alkalinity (d), surface DIC (e) and surface phosphate (f). The global sea surface salinity is also represented in orange on panel (d).

Figure 11: Add a sentence in the caption stating that the timeseries start when the flooding event at that location occurs (if that's the case).

We added "The time series start when land is flooded." in the caption.

References

Deschamps, P., Durand, N., Bard, E., Hamelin, B., Camoin, G., Thomas, A. L., Henderson, G. M., Okuno, J., and Yokoyama, Y.: Ice-sheet collapse and sea-level rise at the Bølling warming 14,600 years ago, Nature, 483, 559–564, 2012.

Harrison, S.: BIOME 6000 DB classified plotfile version 1, available at: http://researchdata.read.

Ivanovic, R. F., Gregoire, L. J., Kageyama, M., Roche, D. M., Valdes, P. J., Burke, A., Drummond, R., Peltier, W. R., and Tarasov, L.: Transient climate simulations of the deglaciation 21–9 thousand years before present (version 1) – PMIP4 Core experiment design and boundary conditions, Geosci. Model Dev., 9, 2563–2587, https://doi.org/10.5194/gmd-9-2563-2016, 2016.

Köhler, P., Nehrbass-Ahles, C., Schmitt, J., Stocker, T. F., and Fischer, H.: A 156 kyr smoothed history of the atmospheric greenhouse gases $CO_2$, $CH_4$, and $N_2O$ and their radiative forcing, Earth Syst. Sci. Data, 9, 363–387, https://doi.org/10.5194/essd-9-363-2017, 2017.

Lin, Y., Hibbert, F. D., Whitehouse, P. L., Woodroffe, S. A., Purcell, A., Shennan, I., and Bradley, S. L.: A reconciled solution of MeltwaterPulse 1A sources using sea-level fingerprinting, Nat. Commun., 12, 1–11, 2021.

Liu, J., Milne, G. A., Kopp, R. E., Clark, P. U., and Shennan, I.: Sea-level constraints on the amplitude and source distribution of Meltwater Pulse 1A, Nat. Geosci., 9, 130–134, 2016.

Nevison, C. D., Manizza, M., Keeling, R. F., Stephens, B. B., Bent, J. D., Dunne, J., Ilyina, T., Long, M., Reslandy, L., Tjiputra, J., and Yukimoto, S.: Evaluating CMIP ocean biogeochemistry and Southern Ocean carbon uptake using atmospheric potential oxygen: Present-day performance and future projection, Geohys. Res. Lett., 43, 2077–2085, https://doi.org/10.1002/2015GL067584, 2016.

Spratt, R. M. and Lisiecki, L. E.: A Late Pleistocene sea level stack, Clim. Past, 12, 1079–1092, https://doi.org/10.5194/cp-12-1079-2016, 2016.

Tarasov, L., Dyke, A. S., Neal, R. M., and Peltier, W. R.: A data-calibrated distribution of deglacial chronologies for the North American ice complex from glaciological modeling, Earth Plan. Sci. Lett., 315, 30–40, https://doi.org/10.1016/j.epsl.2011.09.010, 2012.

---

## Author Response (AR1)

We thank the two reviewers and the editor for the careful reading and critical review of our manuscript. The different comments were very useful to improve the manuscript. The response is organized by first addressing similar comments from both reviewers and the editor on the modelled vegetation at 21 and 15 ka in comparison to pollen data. We then respond to the editor and reviewer 1 comments on the Southern Ocean ventilation and relationship with the $CO_2$ uptake. We finally go through other editor comments and all comments from the two reviewers in separate parts. In the following, the responses to the comments are shown in blue.

**1- Model-data biome comparison**

We acknowledge that the original comparison between the modelled biomes at 15 ka and the pollen data at 21 ka alone was not sufficient. We added a new Fig. 4 to show first the comparison between the modelled biomes and pollen data at 21 ka (a) and then between the modelled biomes at 15 ka and the pollen data at 21 ka (b), since no global synthesis of pollen reconstruction is available for 15 ka yet. A metric method (BNS – Best Neighbour Score) has also been used to quantify the similarity between the model results and the reconstructions.

[Figure]

Figure 4. Biome distribution modelled by JSBACH at 21 ka (a) and 15 ka (b). The superimposed circles are the pollen data from the BIOME6000 Version 4.2 reconstruction at 21 ka for both figures (Harrison et al., 2017). The right plots indicate the best neighbour score, i.e. the similarity between the modelled biomes and the pollen data, for both time period.

The following text has been added in the revised manuscript within the section 3.1 when looking at the land response: "To do so we used the biomisation technique developed in Dallmeyer et al. (2019) to convert the different PFT cover fractions modelled in JSBACH into 9 biomes. We also used the best neighbour score (BNS) metric method presented in Dallmeyer

et al. (2019) to quantify the similarity between the modelled biomes and the pollen data from the BIOME6000 database. This method uses the surrounding grid boxes of the studied grid cell (in each direction of the T31 grid) to compare with the pollen record. The agreement for each record is calculated with the distance weight of the best neighbour in each neighbourhood (using a Gaussian function) and varies between 1 if the modelled biomes in the grid box indicates the same biome as reconstructed and 0 if all grid cells in the neighbourhood disagree with the record. The BNS is the mean of all individual neighbourhood scores. The LGM modelled biomes on Fig. 4a show an overall good agreement with the pollen data with a total BNS value of 0.52. At high latitudes of the Northern Hemisphere, tundra and boreal forests are simulated in regions that are not covered by ice, which is consistent with the few pollen datasets available at these locations (BNS value of 0.78 and 0.19 respectively). Temperate forest is modelled over part of North America, grassland over Europe and temperate/warm forest over East Asia. This is generally in agreement with the pollen record even if some local discrepancies are observed like in central Asia. At low latitudes the model mostly reproduces the tropical forest (over Eastern South America, West Africa and Indonesia) as observed in the pollen data with a BNS value of 0.38 (Figure 4a). Although the LGM conditions were different from those at 15 ka before MWP1a, in absence of other global reconstructions we also used the LGM BIOME6000 pollen record to compare to model results. According to our model, the biome distribution doesn't change much between 21 and 15 ka (Figure 4a,b) so that for many regions, the LGM pollen data show the same pattern as the simulated biomes at 15 ka. The BNS value at 15 ka is similar to the one at 21 ka for tropical forest, warm forest, savanna and desert. However, climatic differences between these two periods lead to small differences between the simulated biomes at 15 ka and the 21 ka pollen data which explains the lower total BNS value of 0.45 compared to 0.52 (Figure 4a,b). Part of the LGM tundra at high latitudes of the Northern Hemisphere is replaced by the boreal forest or grassland at 15 ka. At low latitudes, there is a slightly larger extent of the temperate forest over East Asia and of the tropical forest over South America at 15 ka. The tropical forest over Indonesia is however already present since the LGM."

**2- Oceanic ventilation and $CO_2$ uptake**

Following comment from the editor and reviewer 1 on the relationship between oceanic circulation in the Southern Ocean and $CO_2$ uptake, we changed the text to make it clearer in the revised manuscript. In general, MPI-ESM tends to have a higher anthropogenic $CO_2$ uptake in the Southern Ocean compared to other CMIP models (Nevison et al., 2016) linked to an interplay of ventilation, biological production and deep/intermediate water formation.

We added the following text in the discussion section of the revised manuscript: "In our model, the physical state of the ocean, and in particular the AMOC and the ventilation of the Southern Ocean, show only little variation before 15 ka. Thus, the global net air-sea $CO_2$ flux is generally close to zero until 17.3 ka (Figure 2e), and then becomes mostly negative, i.e. the global ocean becomes a carbon sink, for several millennia due to the prescribed rising atmospheric $CO_2$ mixing ratio (Figure 2d). We do not find an enhanced outgassing of $CO_2$ in the Southern Ocean due to an increased ventilation in our model. Sensitivity studies with an Earth System Model of Intermediate Complexity suggested that the observed atmospheric $CO_2$ increase after 17.3 ka could be attributed to an enhanced formation of Antarctic intermediate and/or deep water

due to decreased buoyancy forces and/or changes in the westerlies in the Southern Hemisphere (Menviel et al., 2018)."

**3- Response to editor's comments**

Here I do not think it is adequate to simply remove the reference to proxy data on AMOC changes, as it is entirely apposite that the model response appears to run contrary to what proxy reconstructions are generally interpreted to show; i.e. a strong resumption of the AMOC around MWP1a, and not a collapse at all. This is an important point that I think is best brought out clearly, and discussed (e.g. is it expected that differences between model runs with/without OM are not affected by differences in the AMOC state and its variability?).

Based on your comment we specified that the Pa/Th maximum is observed before the beginning of MWP1a, between 17.5 and 15 ka. However, the AMOC weakening in the model starts about 2500 year later and its duration is significantly shorter compared to that suggested by the proxy data. The timing of the simulated AMOC weakening is mainly regulated by temporal variations in the volume of the prescribed ice sheet reconstruction. In our model, the ice sheet volume decrease is considered as liquid meltwater discharge, ignoring the discharge of icebergs which would lead to slower freshwater input to the ocean. Thus, a pulse-like meltwater occurs during 15-14 ka, leading to a rapid AMOC weakening.

In addition, I would add that the AMOC value of 23 Sv, referenced in line 215 (from Muglia & Schmittner, 2015) might arguably be superseded by their later estimate based on a fit to existing nutrient and carbon isotope data (Muglia et al., 2018), which is much weaker. I would invite you to consider whether this is worth noting, and whether or not you feel it could be useful to discuss the potential implications (if any) for your results if the background ocean state and its perturbations (e.g. see above) turned out to be inaccurate. For example, do you expect inferences that depend on the offsets between model runs with/without OM implemented to be influenced by the AMOC state and its variability. A comment on this might help to clarify things for concerned readers.

A weaker AMOC during the LGM as proposed in Muglia et al. (2018) could have implications for the climate and ocean physics/biogeochemistry, like an increase in carbon accumulation in deep and bottom waters. However, in our model, the physical state of the ocean, and in particular the AMOC and the ventilation of the Southern Ocean, show only little variation before 15 ka. The outcomes of local $CO_2$ outgassing during MWP1a originate from the terrestrial organic matter fluxes that happen between the land and the ocean during flooding events. These flooding events depend on the sea level change over MWP1a, induced by freshwater inputs from the prescribed GLAC1D ice sheet reconstruction. Still, we added the reference of Muglia et al. (2018) in the main text.

**4- Point-by-point response to reviewer 1**

1) Title: I suggest "Local oceanic CO2 outgassing…."

We agree that this suggested title is indeed more appropriate and highlights the more local impact of the investigated processes. We changed the title of the manuscript to "Local oceanic CO$_2$ outgassing triggered by terrestrial carbon fluxes during deglacial flooding".

2) Introduction:

The structure of the introduction needs to be improved and some aspects that are the focus of the manuscript are currently missing. The introduction does not properly mention the current hypotheses to explain glacial/interglacial changes in pCO$_2$. L52-53 is confusing, as there has been a lot of work done in understanding glacial-interglacial changes in atmospheric CO$_2$, including some transient simulations. On the other hand, the introduction includes one paragraph on the impact of Heinrich events on the carbon cycle from a modelling perspective, but links with the current studies are not made. The introduction should also include a paragraph on estimates of glacial-interglacial changes in terrestrial carbon to provide a perspective on the modelling outputs. Finally, since MWP1A is mentioned throughout the manuscript, a brief paragraph on MWP1a should be added. This paragraph could describe estimates of the timing of MWP1A, its magnitude and the potential origin of this meltwater pulse (NH vs SH).

Thank you for pointing out these aspects. We revised the introduction with a more detailed section on glacial/interglacial CO$_2$ variations and on the processes of the ocean physics and biogeochemistry that can explain these CO$_2$ variations. We also added a new section focusing on MWP1a. The text of the introduction has been modified in the revised manuscript.

3) Deglacial sea-level rise

Since the results of the present study are dependent on the sea-level rise, a timeseries of the simulated deglacial sea-level rise along with paleo-estimates should be included in Figure 2. The implications of potential differences between simulated and estimated deglacial sea-level rise should be discussed.

We added two curves in Fig. 2 (see panel b in the new figure below). The first one is the global sea level stack from Spratt and Lisiecki (2016) based on isotopic measurements. The second one is the modelled global sea level change based on the freshwater inputs to the ocean that are derived accordingly to the ice sheet volume change from the GLAC1D ice sheet reconstruction from Tarasov et al. (2012).

The global sea level change in the model increases by 67.4 m between 21-12 ka, which is close to the value of 69 m obtained from proxy data (Spratt and Lisiecki, 2016). During MWP1a, the global sea level change in the model shows quantitative differences compared to Spratt and Lisiecki (2016) record. Uncertainties exist in the prescribed ice sheet reconstructions that could explain such difference. For instance, the ice sheet volume and the timing of freshwater input show noticeable differences between GLAC1D and ICE6G reconstructions (see Ivanovic et al. 2016 for a comparison). The global sea level increases of 19.6 m for the 500 years of largest freshwater inputs in the model. This is in the high range of the previous estimations with a global sea level increase from 8.6 to 20.2 m (Deschamps et al., 2012; Liu et al., 2016;

Lin et al., 2021). Then, between 14-12 ka, the sea level in the model only slightly increases in comparison to the Spratt and Lisiecki (2016) record. We added this discussion in the main text of the revised manuscript.

[Figure]

[Figure]

Figure 2. Time series of land, ocean and atmosphere variables over the last deglaciation. The presented outputs start at 21 ka. (a) Freshwater input to the global ocean. (b) Global sea level estimate from Spratt and Lisiecki (2016) (light purple) and modelled in MPI-ESM based on the freshwater inputs (dark purple). (c) Atlantic Meridional Overturning Circulation streamfunction. (d) $CO_2$ concentration measured in ice cores (Köhler et al., 2017). (e) Modelled global net $CO_2$ flux between the ocean and the atmosphere. Positive $CO_2$ flux mean that the ocean is outgassing to the atmosphere and negative $CO_2$ flux mean that the ocean is uptaking carbon. (f) Global ocean net primary production. (g) Total carbon in all terrestrial carbon pools, i.e. vegetation, soil and litter. The thick darker curves are 500 years running mean for the panel (a) and 50 years running mean for the panels (c), (e) and (f). A zoom over MWP1a is presented on the right.

4) Line by line comments

P1, L. 1: The first sentence of the abstract is odd. It needs to be rephrased, and most likely split in two sentences.

We rephrased the first sentence to: "Exchanges of carbon between the ocean and the atmosphere are key processes that influence past climates via glacial/interglacial variations of the $CO_2$ concentration".

P1, L.2: I suggest "induces a sea-level rise"

Change done.

P1, L. 10: I suggest "leads to 21.2 GtC transfer of terrestrial organic carbon to the ocean"

Done.

P1, L. 20: "including" instead of "triggered"

Done.

P2, L. 28: Consider adding references to Lambeck et al. 2014.

We added this reference to the main text.

P7, L. 190: Weren't the nutrient concentrations adjusted for the lower sea-level at the LGM?

We initialized the model with nutrient concentrations from a present day MPI-ESM simulation. We didn't adjust for the 3.5 % change in oceanic volume between the present day and the LGM, i.e. the prescribed total nutrient inventory at LGM is slightly smaller than for Preindustrial. We specify this aspect in the main text of the manuscript.

In addition, the ocean to atm. $CO_2$ flux shown in Fig. 2c is not really explained, and barely mentioned in the text. However, given the experimental setup, it might be simply responding to the forced changes in atmospheric $CO_2$, and to changes in surface solubility.

Yes, you are correct. We added the following text in the revised manuscript: "As we prescribe the atmospheric $CO_2$ concentration, we omit any interaction between the land and the marine carbon cycle. Thus, the air-sea exchange is primarily following the atmospheric $CO_2$ increase, modified locally by physically induced changes of the circulation, biogeochemistry and surface solubility."

A timeseries of atmospheric $CO_2$ should be included in figure 2 to better understand the ocean-atm $CO_2$ flux (2c), and maybe a timeseries of globally averaged SST.

Following this comment, we added the atmospheric $CO_2$ concentration measured in ice cores (Köhler et al., 2017) and prescribed in the model on the panel (d) in the new Fig. 2 (see figure above).

P9, L. 230: The Pa/Th record from the Bermuda rise suggests an AMOC weakening during HS1, but not necessarily during MWP1a. I think that the most recent chronology suggest the end of HS1 and thus beginning of Bolling Allerod at 14.6 ka, contemporary with the beginning of MWP1a.

Yes, this is correct, the Pa/Th maximum is observed before the beginning of MWP1a, between 17.5 and 15 ka. The AMOC weakening in the model starts about 2500 year later and its duration is significantly shorter compared to that suggested by the proxy data. The timing of the simulated AMOC weakening is mainly regulated by temporal variations in the volume of the prescribed ice sheet reconstruction. In our model, the ice sheet volume decrease is considered as liquid meltwater discharge, ignoring the discharge of icebergs which would lead to slower freshwater input to the ocean. Thus, a pulse-like meltwater occurs during 15-14 ka, leading to a rapid AMOC weakening.

P12, L. 283 and throughout: I understand why you are referring to "terrestrial organic carbon input to the atmosphere", however this is not correct and could be confusing. It might be better to simply refer to "terrestrial carbon input to the atmosphere".

Thank you for addressing this issue on the terminology, which is indeed confusing. The correct wording must be emission of carbon origin from remineralization of short-living terrestrial organic matter. We replaced "terrestrial organic carbon input to the atmosphere" in the manuscript by "flooding induced terrestrial carbon emissions".

P19, L. 369 and throughout manuscript: I am not sure that the use of North/central and south Indonesia is correct. Maybe it is more appropriate to refer to the Sunda and Sahul shelves.

We refrained here from using the terms "Sunda and Sahul shelves" or the name of islands (e.g. Borneo, Sumatra…) because the resolution of the model is too coarse to simulate the shelves or the individual small islands in Indonesia. Instead, we defined an equatorial box between 15°N and 15°S around Indonesia where the largest differences are observed, and divide this box in three parts: northern part, central part and southern part (see new Fig. 10). These three parts are then used for the discussion in the main text. Figure 9 has also been revised accordingly.

[Figure]

Figure 10. Anomaly of the mean surface $CO_2$ flux between the simulations with and without terrestrial organic matter inputs to the ocean averaged over MWP1a. Negative values indicate flux from the atmosphere to the ocean (uptaking) and positive values indicate flux from the ocean to the atmosphere (outgassing). An equatorial box is defined between 15°N and 15°S and subdivided in three areas: northern part, central part and southern part.

Figure 8: Add AMOC and/or meltwater timeseries?

We prefer not to add the AMOC or meltwater timeseries on the Fig. 8 since they are already shown in previous figures. Instead, we added on the panel (d) in the new Fig. 8 the global sea surface salinity evolution during MWP1a since changes in surface alkalinity are mainly controlled by changes in sea surface salinity, which are induced by freshwater inputs at that time. We also removed the indication of the timing of flooding events on the right figures and only mention them in the main text.

[Figure]

Figure 8. Anomaly of the mean surface alkalinity (a), surface dissolved inorganic carbon (b), and surface phosphate (c) between the simulations with and without terrestrial organic matter fluxes averaged over MWP1a. Time evolution of the two simulations during MWP1a for annual mean global surface alkalinity (d), surface DIC (e) and surface phosphate (f). The global sea surface salinity is also represented in orange on panel (d).

Figure 11: Add a sentence in the caption stating that the timeseries start when the flooding event at that location occurs (if that's the case).

We added "The time series start when land is flooded." in the caption.

**5- Point-by-point response to reviewer 2**

1) Introduction:

Maybe expand the section on glacial-interglacial $CO_2$ variations (p.2, ll.39-49). In the context of the current study, recent estimates of total changes in land carbon storage between the last glacial maximum (LGM) and preindustrial (PI) might be of interest (e.g. Müller and Joos, 2020 BG; Jeltsch-Thömmes et al., 2019 CP). Further, many studies have invoked processes other than physical changes in the ocean (see e.g. Menviel et al., 2012 QSR or Sigman et al., 2010 Nature for a review, and many others) to explain glacial-interglacial $CO_2$ variations.

Following your comment, we further developed the section on the glacial/interglacial atmospheric $CO_2$ variations to explain them from changes in physical and biological conditions of the ocean. We also gave an estimate of the terrestrial and marine carbon pools for the LGM and Preindustrial. The text of the introduction has been modified in the revised manuscript.

2) Prescribed atmospheric $CO_2$ concentration:

At the end of section 2.1 the authors note that all atmospheric concentrations are prescribed in the simulations. One of the goals of glacial-interglacial simulations with ESMs is to simulate the change in atmospheric $CO_2$ concentration, i.e. the ~90 ppm increase since the LGM. While making sure to have the correct atmospheric inventory, prescribing atm. $CO_2$ comes with drawbacks. For example, without other changes this would lead to a smaller LGM DIC inventory as the atmosphere would act as a sink until equilibrium is reached with the ocean. The authors circumvent this by initializing the spinup simulations with higher alkalinity concentrations. Is there a specific reason for not letting atm. $CO_2$ evolves freely over the course of the simulation? I don't think, though, this would change the findings of the study, as the effect of terrestrial organic carbon fluxes is diagnosed from the difference of two runs, but would like to see at least a short discussion of this choice.

Up to now, we only ran the deglaciation simulation with prescribed $CO_2$ to test and validate the state of the model during the deglaciation with the new developments that have been added in the model framework. A new simulation with prognostic atmospheric $CO_2$ / interactive carbon cycle (i.e. prognostic $CO_2$ for global carbon cycle but prescribed $CO_2$ for radiation) will be performed in near future. This will allow us to address the gap on the interaction between the ocean biogeochemistry and the climate during the last deglaciation. We added a new sentence to explain it in Section 2.4.

Are changes in tracer concentrations as a result of lower sea-level considered here as well?

We initialized the model with nutrient concentrations from a present day MPI-ESM simulation. We didn't adjust for the 3.5 % change in oceanic volume between the present day and the LGM, i.e. the prescribed total nutrient inventory at LGM is slightly smaller than for Preindustrial. We specify it in the revised version of the manuscript.

3) Simulated terrestrial carbon inventory

In general, I was a bit surprised to read that the effect of terrestrial organic carbon fluxes as a result of flooding are rather small and am wondering whether this might link to the size of the simulated terrestrial carbon inventory and thus the amount of carbon available in flooded

gridcells. On page 10, l.232-233 the authors state that the terrestrial carbon inventory increased from 922.9 GtC to 1302.7 GtC between 21-15 kaBP and amounts to 1563.6 GtC in 12 kaBP. I am no expert on land modeling, but in a recent paper Müller and Joos (2020, BG) simulate total terrestrial carbon at the LGM at about 2000 GtC, which increases to about 2500 GtC in 12 kaBP. This is almost twice the amount shown in this study. Also, Ganopolski and Brovkin (2017, CP) simulate a larger terrestrial carbon inventory. Is the assumption correct that a higher terrestrial carbon pool would also increase the terrestrial organic carbon flux during flooding? If yes, this might be a point to be included in the discussion of uncertainties of the findings.

The change in terrestrial carbon content from 21 to 12 ka is slightly higher in our model with 640 GtC compared to 500 GtC from the study of Müller and Joos (2020). However, it is true that the initial value for the LGM is larger in their paper than in our simulation. This could partly be explained by the fact that we don't include peatland in this version of the model, so that we miss around 300 GtC as estimated in their paper. Other estimations (e.g. Prentice et al., 2011) suggest a total land carbon content of 1070 GtC for the LGM using a Dynamic Global Vegetation Model, which is close to our value. Compiled land carbon estimations from different modelling studies also suggest a change between the LGM and the preindustrial from 450 to 1250 GtC (Jeltsch-Thömmes et al., 2019). Even if our simulation doesn't go yet further than 12 ka, we are within the range of the estimated change.

More carbon stored in the terrestrial carbon pools doesn't necessarily imply an increase in terrestrial organic carbon fluxes to the ocean during flooding. The local carbon pool calculated as the sum of the vegetation, litter and soils pools varies depending on the vegetation type and on the latitude. During the simulated MWP1a, the flooding induced terrestrial organic carbon fluxes mainly happen at low latitudes, characterized by tropical vegetation which has a relatively high carbon biomass above ground (short-living material such as leaves) and typically doesn't have a high carbon content in soils. As in our simulation only long-living material from land carbon pools enters the ocean during a flooding event, our results for MWP1a might rather be insensitive to a higher local total carbon content in the tropical vegetation. Of course, this could be different for high latitudes characterized by organic rich soils in tundra, shrub or grassland which could be flooded later in the deglaciation.

In the same paragraph (p.10, ll.235-237) include Müller and Joos, 2020 BG into the estimates of terrestrial carbon evolution.

We added the reference in the text.

Are peatlands included in the land component of the model?

The applied version of JSBACH does not include peatlands.

Are there other uncertainties that would be good to be discussed (other than C:N:P ratios)?

This is a good point to mention. Besides the C:N:P ratios, the remineralization rates of the terrestrial organic matter in sea water are not well constrained parameters. The choice of different rates could lead to higher or lower $CO_2$ flux to the atmosphere. In the deglaciation run and presented sensitivity simulations with higher and lower stoichiometries of terrestrial organic matter, the remineralization rates were prescribed to $2.7 \times 10^{-5}$ $d^{-1}$ for wood, $2.7 \times 10^{-4}$ $d^{-1}$ for woody litter and $5.5 \times 10^{-4}$ $d^{-1}$ for humus. To investigate the influence of higher

remineralization rates, we perform an additional sensitivity experiment for MWP1a. The new values are $1.0 \times 10^{-4}$ d$^{-1}$ for wood, $2.0 \times 10^{-3}$ d$^{-1}$ for woody litter and $8.0 \times 10^{-3}$ d$^{-1}$ for humus. This simulation uses the same higher stoichiometry ratios as one of the first sensitivity studies (see Table 3 in the manuscript) to get an upper estimate of the potential impact of terrestrial fluxes.

We observe higher $CO_2$ outgassing in the defined equatorial box over a shorter time period. Figure 11 has been revised including a new grey curve that represents the simulation with high stoichiometry and high remineralization rates. For the northern part, the $CO_2$ flux to the atmosphere reaches $20 \times 10^{-9}$ kg C m$^{-2}$ s$^{-1}$ after the flooding at 14.64 ka and decrease twice as fast as the simulation with high stoichiometry only (Figure 11a). Similar behaviour is observed for central and southern part with an outgassing peak after the flooding at 14.54 ka and 14.18 ka of $32 \times 10^{-9}$ kg C m$^{-2}$ s$^{-1}$ and $27 \times 10^{-9}$ kg C m$^{-2}$ s$^{-1}$ respectively (Figure 11b, c). This increased $CO_2$ flux to the atmosphere is primarily a result of wood remineralization since, as for previous simulations, wood dominates the terrestrial organic matter input to the ocean during flooding events at that latitude. Since wood is not buried in the sediment, the amount of material that can be remineralized is the same as in previous simulation, but at faster rate. Part of the outgassing is still due to the remineralization of woody litter and humus before they are buried. This text has been added at the end of Section 3.3 of the revised manuscript.

[Figure]

Figure 11. Evolution of the surface $CO_2$ flux during MWP1a for flooded grid cells in northern part (a), central part (b) and southern part (c) of the equatorial box defined between 15°N and 15°S for 5 different simulations: the reference simulation with the terrestrial organic matter fluxes (dark blue), the sensitivity experiment without terrestrial organic matter fluxes (light blue), the sensitivity experiment with low stoichiometry for terrestrial organic matter (red), the sensitivity experiment with high stoichiometry (orange) and the sensitivity experiment with high stoichiometry and high remineralization rates (grey). 50 years running mean is plotted for each simulation. Positive values indicate an outgassing to the atmosphere and negative values indicate an uptake by the ocean. The time series start when a flooding event occurs at that location.

p.5, ll.129-130: either 'presented a new development' or 'presented new developments'

Change done.

**References**

Deschamps, P., Durand, N., Bard, E., Hamelin, B., Camoin, G., Thomas, A. L., Henderson, G. M., Okuno, J., and Yokoyama, Y.: Ice-sheet collapse and sea-level rise at the Bølling warming 14,600 years ago, Nature, 483, 559–564, 2012.

Harrison, S.: BIOME 6000 DB classified plotfile version 1, available at: http://researchdata.read.

Ivanovic, R. F., Gregoire, L. J., Kageyama, M., Roche, D. M., Valdes, P. J., Burke, A., Drummond, R., Peltier, W. R., and Tarasov, L.: Transient climate simulations of the deglaciation 21–9 thousand years before present (version 1) – PMIP4 Core experiment design and boundary conditions, Geosci. Model Dev., 9, 2563–2587, https://doi.org/10.5194/gmd-9-2563-2016, 2016.

Jeltsch-Thömmes, A., Battaglia, G., Cartapanis, O., Jaccard, S. L., and Joos, F.: Low terrestrial carbon storage at the Last Glacial Maximum: constraints from multi-proxy data, Clim. Past, 15, 849–879, https://doi.org/10.5194/cp-15-849-2019, 2019.

Köhler, P., Nehrbass-Ahles, C., Schmitt, J., Stocker, T. F., and Fischer, H.: A 156 kyr smoothed history of the atmospheric greenhouse gases $CO_2$, $CH_4$, and $N_2O$ and their radiative forcing, Earth Syst. Sci. Data, 9, 363–387, https://doi.org/10.5194/essd-9-363-2017, 2017.

Lin, Y., Hibbert, F. D., Whitehouse, P. L., Woodroffe, S. A., Purcell, A., Shennan, I., and Bradley, S. L.: A reconciled solution of MeltwaterPulse 1A sources using sea-level fingerprinting, Nat. Commun., 12, 1–11, 2021.

Liu, J., Milne, G. A., Kopp, R. E., Clark, P. U., and Shennan, I.: Sea-level constraints on the amplitude and source distribution of Meltwater Pulse 1A, Nat. Geosci., 9, 130–134, 2016.

Menviel, L., Spence, P., Yu, J., Chamberlain, M. A., Matear, R. J., Meissner, K. J., and England, M. H.: Southern Hemisphere westerlies as a driver of the early deglacial atmospheric $CO_2$ rise, Nat. Commun., 9, 2503, https://doi.org/10.1038/s41467-018-04876-4, 2018.

Müller, J. and Joos, F.: Global peatland area and carbon dynamics from the Last Glacial Maximum to the present–a process-based model investigation, Biogeosciences, 17, 5285–5.

Nevison, C. D., Manizza, M., Keeling, R. F., Stephens, B. B., Bent, J. D., Dunne, J., Ilyina, T., Long, M., Reslandy, L., Tjiputra, J., and Yukimoto, S.: Evaluating CMIP ocean biogeochemistry and Southern Ocean carbon uptake using atmospheric potential oxygen: Present-day performance and future projection, Geohys. Res. Lett., 43, 2077–2085, https://doi.org/10.1002/2015GL067584, 2016.

Prentice, I. C., Harrison, S. P., and Bartlein, P. J.: Global vegetation and terrestrial carbon cycle changes after the last ice age, New Phytologist, 189, 988–998, https://doi.org/10.1111/j.1469-8137.2010.03620.x, 2011.

Spratt, R. M. and Lisiecki, L. E.: A Late Pleistocene sea level stack, Clim. Past, 12, 1079–1092, https://doi.org/10.5194/cp-12-1079-2016, 2016.

Tarasov, L., Dyke, A. S., Neal, R. M., and Peltier, W. R.: A data-calibrated distribution of deglacial chronologies for the North American ice complex from glaciological modeling, Earth Plan. Sci. Lett., 315, 30–40, https://doi.org/10.1016/j.epsl.2011.09.010, 2012.

---

## Author Response (AR2)

Dear Editor, we would like to thank you for pointing out this aspect of the manuscript that needed to be clarified. We respond below (in purple) to your comments and made corresponding changes in the revised manuscript.

Thank you for submitting your revised manuscript. After reviewing the changes that have been made, I am afraid that I find the additions made with respect to the observed versus modelled AMOC changes at the time of MWP1a rather misleading. It is not the case that "a decrease in AMOC strength [associated with MWP1a] is also deduced from 231Pa/230Th data". Rather the opposite is observed: proxy data suggest that the AMOC *strengthens* instead. It is very difficult to see this as merely a question of 'timing' (i.e. the same thing occurring 2500 years later in the model), as the decrease in AMOC strength reconstructed by McManus et al (2004), and coherent with a host of other proxy data, coincides broadly with the onset of Heinrich-stadial 1 (HS1), and not the onset of the Bolling-Allerod and MWP1a. The difference in timing between MWP1a and the onset of HS1 is generally thought to be quite well constrained (at least to within 2500 years).

In light of these issues, I still find that there is a need to address the clear discrepancy between the modelled AMOC effects, and observations (i.e. the most widespread and consistent interpretation of a great deal of proxy data, including Pa/Th, as well as radiocarbon, stable isotopes, oxygenation proxy data). As described before, this discrepancy does not necessarily invalidate interpretations of the differences between model runs with and without organic carbon mobilisation implemented. However, the mismatch and its implications do need to be a accurately noted and discussed. This may require that you delve further into the literature on deglacial chronostratigraphy, U/Th dating of sea level changes, and AMOC/deep-water hydrographic reconstructions. (The timing of WMP1a with respect to reconstructed changes in the AMOC is a long-standing puzzle.)

We first modified the text about the model-data comparison of the LGM AMOC strength, starting line 245 on the new version of the manuscript:

"The LGM AMOC strength is 22.5 Sv (1 Sv = $10^6$ m$^3$ s$^{-1}$) which is within the range of a multimodel mean LGM maximum AMOC value of 23 ± 3 Sv (Muglia and Schmittner, 2015), even if it is still unclear whether the AMOC was weaker or stronger during the LGM than in the Preindustrial. One data assimilation study supports a strong AMOC during the LGM with a value of 21.3 Sv (Kurahashi-Nakamura et al., 2017). In contrast, a recent estimate based on modelling experiments constrained by isotopic data suggests a weaker AMOC during the LGM, with values between 6 and 9 Sv (Muglia et al., 2018)."

We also corrected and extended the discussion on the modelled AMOC strength and variation in comparison to data. The text below replaces the existing text in the discussion, starting line 268 on the revised manuscript:

"The variability of the simulated AMOC is mainly regulated by temporal changes in the volume of the prescribed ice sheet reconstruction. In our model, the ice sheet volume decrease is considered as liquid meltwater discharge, ignoring the discharge of icebergs. Freshwater inputs deduced from the GLAC-1D ice sheet reconstruction show low variations during the LGM and only a slight increase during the Heinrich Stadial 1. Thus, we can't expect pronounced AMOC changes during the period of Heinrich Stadial 1 in the model. Between 15-

14 ka, we simulate a decrease of the AMOC strength following massive freshwater inputs in the Northern Hemisphere. This period of MWP1a is also characterized by a rapid sea level increase, which is recorded in radiocarbon dates from the Sunda Shelf and U/Th measurements on corals offshore from Tahiti, confirming a timing of MWP1a between 14.65 to 14.31 ka (Hanebuth et al., 2000, Deschamps et al., 2012). However, the temporal variation of the AMOC strength estimated from $^{231}$Pa/$^{230}$Th tends to show already a decrease between 17.5-15 ka, i.e. before the MWP1a, and an increase back to a high value between 15-14 ka (McManus et al., 2004). To achieve a good agreement between simulated and proxy-data derived AMOC variations, He (2011) showed in a modelling study the necessity of meltwater fluxes from Antarctica during MWP1a. Other hosing experiments also emphasize the sensitivity of the oceanic circulation, and thus the AMOC, on the location of the freshwater input (e.g. Smith and Gregory, 2009; Menviel et al., 2011). As previously discussed, the meltwater input deduced from GLAD-1D is located primarily in the Northern Hemisphere, which might explain the different temporal evolution of the simulated AMOC. In the following, we refer to MWP1a as the period of rapid sea level change due to large freshwater inputs to the ocean to evaluate the effect of land-sea exchanges during flooding events."

**References**

Deschamps, P., Durand, N., Bard, E., Hamelin, B., Camoin, G., Thomas, A. L., Henderson, G. M., Okuno, J., and Yokoyama, Y.: Ice-sheet collapse and sea-level rise at the Bølling warming 14, 600 years ago, Nature, 483, 559–564, https://doi.org/10.1038/nature10902, 2012.

Hanebuth, T., Stattegger, K., and Grootes, P. M.: Rapid Flooding of the Sunda Shelf: A Late-Glacial Sea-Level Record, Science, 288, 1033–1035, https://doi.org/10.1126/science.288.5468.1033, 2000.

He, F.: Simulating transient climate evolution of the last deglaciation with CCSM 3, PhD, Atmospheric and Ocean Sciences, University of Wisconsin-Madison, 185 pp., 2011.

Kurahashi-Nakamura, T., Paul, A., and Losch, M.: Dynamical reconstruction of the global ocean state during the Last Glacial Maximum, Paleoceanography, 32, 326–350, https://doi.org/10.1002/2016pa003001, 2017.

McManus, J. F., Francois, R., Gherardi, J.-M., Keigwin, L. D., and Brown-Leger, S.: Collapse and rapid resumption of Atlantic meridional circulation linked to deglacial climate changes, Nature, 428, 834–837, https://doi.org/10.1038/nature02494, 2004.

Menviel, L., Timmermann, A., Timm, O. E., and Mouchet, A.: Deconstructing the Last Glacial termination: the role of millennial and orbital-scale forcings, Quaternary Sci. Rev., 30, 1155–1172, https://doi.org/10.1016/j.quascirev.2011.02.005, 2011.

Muglia, J. and Schmittner, A.: Glacial Atlantic overturning increased by wind stress in climate models, Geophys. Res. Lett., 42, 9862–9868, *https*://doi.org/10.1002/2015GL064583, 2015.

Muglia, J., Skinner, L. C., and Schmittner A.: Weak overturning circulation and high Southern Ocean nutrient utilization maximized glacial ocean carbon, Earth Planet Sc. Lett., 496, 47–56, https://doi.org/10.1016/j.epsl.2018.05.038, 2018.

Smith, R. S. and Gregory, J. M.: A study of the sensitivity of ocean overturning circulation and climate to freshwater input in different regions of the North Atlantic, Geophys. Res. Lett., 36, L15701, doi:10.1029/2009gl038607, 2009.